# MLP MEMORY: A RETRIEVER-PRETRAINED MEMORY FOR LARGE LANGUAGE MODELS

**Rubin Wei**[1*] **Jiaqi Cao**[1,4*] **Jiarui Wang**[1] **Jushi Kai**[1] **Qipeng Guo**[2]
**Bowen Zhou**[2,3] **Zhouhan Lin**[1,2†]
[1]LUMIA Lab, Shanghai Jiao Tong University  [2]Shanghai Artificial Intelligence Laboratory
[3]Electronic Engineering, Tsinghua University  [4]SJTU Paris Elite Institute of Technology
weirubinn@gmail.com, lin.zhouhan@gmail.com
Code & model available at https://github.com/LUMIA-Group/MLPMemory

## ABSTRACT

Modern approaches to enhancing Large Language Models' factual accuracy and knowledge utilization face a fundamental trade-off: non-parametric retrieval-augmented generation (RAG) provides flexible access to external knowledge but suffers from high inference latency and shallow integration, while parametric fine-tuning methods like LoRA risk catastrophic forgetting and degraded general capabilities. In this work, we propose MLP Memory, a lightweight parametric module that learns to internalize retrieval patterns without explicit document access. By pretraining an MLP to imitate a $k$NN retriever's behavior on the entire pretraining dataset, we create a differentiable memory component that captures the benefits of retrieval-based knowledge access in a fully parametric form. Our architecture integrates this pretrained MLP Memory with Transformer decoders through simple probability interpolation, achieving 12.3% relative improvement on five question-answering benchmarks and 5.2 points absolute gain across nine general NLP tasks, while reducing hallucinations by up to 10 points on HaluEval. Moreover, MLP Memory delivers 2.5× faster inference than RAG with superior accuracy. Our findings show that learning retrieval patterns parametrically bridges the gap between efficient inference and effective knowledge access, offering a practical alternative to both RAG and fine-tuning approaches.

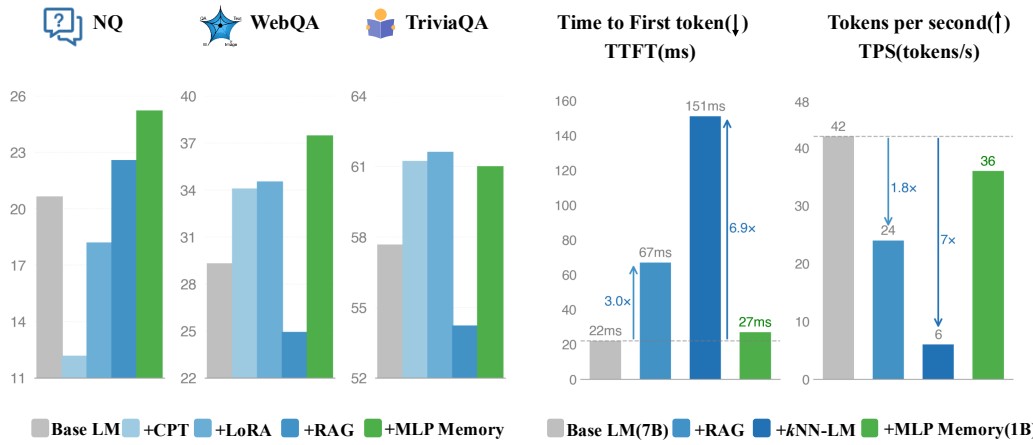

Figure 1: Performance and efficiency comparison. **Left**: accuracy across three QA benchmarks. MLP Memory consistently outperforms the base model, surpassing both parametric methods (CPT, LoRA) and non-parametric retrieval (RAG). **Right**: inference efficiency, measured by time to first token (TTFT, ↓ lower is better) and tokens per second (TPS, ↑ higher is better). RAG results are shown for top-5 retrieval. $k$NN-LM is accelerated via dimension reduction (4096→256), and both RAG and $k$NN-LM use the Wikipedia-2021 retrieval corpus. MLP Memory uses 1B parameters.

---

*Equal contribution.

†Zhouhan Lin is the corresponding author.

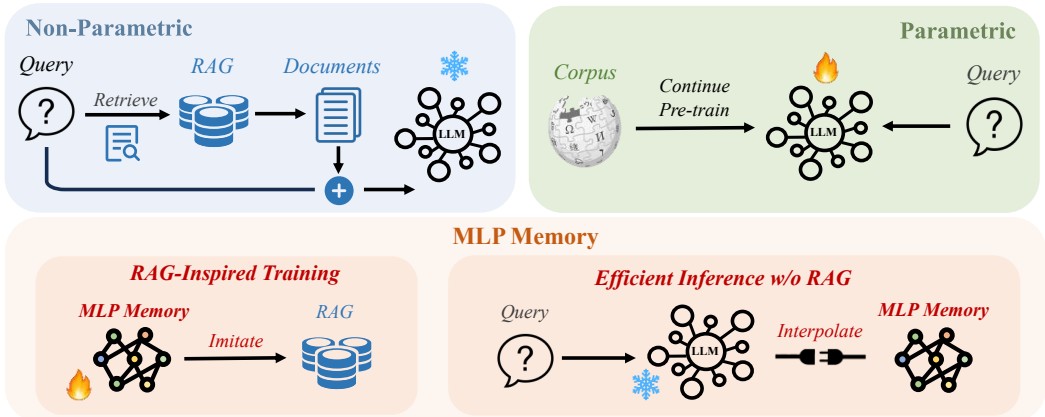

Figure 2: Approaches to enhance factual accuracy and knowledge utilization. Top left: Non-parametric RAG provides flexible knowledge access but suffers from high latency. Top right: Parametric fine-tuning risks catastrophic forgetting. Bottom: MLP Memory learns retrieval patterns during training (left) and enables efficient inference without explicit retrieval (right).

# 1 INTRODUCTION

Decoder-only architectures such as GPT (Brown et al., 2020), LLaMA (Grattafiori et al., 2024), Qwen (Qwen et al., 2025), and DeepSeek (Liu et al., 2024) have achieved remarkable success in various tasks, including open-ended text generation (OpenAI et al., 2024), code completion (Chen et al., 2021), image synthesis (Chen et al., 2020), and multimodal reasoning (Liu et al., 2023). However, despite their impressive capabilities, these models often struggle with effective knowledge utilization, producing responses that may be fluent but fail to accurately leverage the factual information encoded in their parameters.

Current approaches to enhance knowledge utilization in LLMs face significant trade-offs. Retrieval-augmented generation (RAG) methods (Lewis et al., 2021; Peng et al., 2023; Gao et al., 2022; Izacard et al., 2022) dynamically fetch relevant documents to ground model outputs, providing flexible access to external knowledge sources. However, these non-parametric approaches introduce substantial inference latency through expensive nearest-neighbor searches and longer context from retrieved documents. They also suffer from shallow integration with the base model, as the retrieval component remains isolated from the LLM's computational graph. Conversely, parametric adaptation methods such as continued pre-training (CPT) and LoRA (Hu et al., 2022) directly modify model weights to incorporate domain-specific knowledge. While computationally efficient at inference time, these approaches risk catastrophic forgetting of previously learned capabilities and often degrade performance on general tasks, requiring careful task-specific tuning that limits their broader applicability. Figure 2 illustrates how our approach differs fundamentally from both non-parametric retrieval methods and parametric adaptation approaches.

In contrast to decoder-only LLMs, neuroscience research reveals a lateralized human brain where language processing is dominated by the left hemisphere while memory formation occurs in the hippocampus (Gazzaniga, 2005b;a; Douglas, 1967). This insight has inspired memory-augmented models in machine learning. Early approaches like Memory Networks (Weston et al., 2015) enabled read/write operations on external memory, while Sparse Access Memory introduced differentiable memory access schemes. However, these were task-specific with limited general applicability. In the LLM era, methods such as Memory Transformers (Burtsev et al., 2021) incorporate trainable memory tokens for global context, while AutoCompressors (Chevalier et al., 2023) compress long contexts into summary vectors. Nevertheless, these memory tokens primarily function as working memory supplements for context extension rather than long-term memory capable of retaining information from the entire training corpus.

In this work, we propose an external memory for LLM that is pretrained to mimic a retriever on the entire pretraining dataset. Specifically, following the RAG setting in $k$NN-LM (Khandelwal et al.,

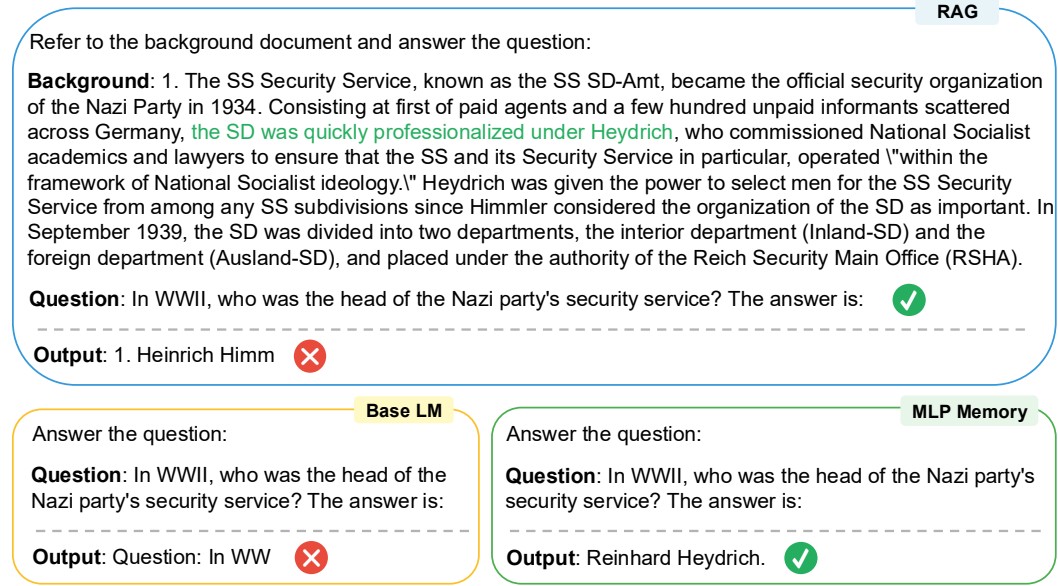

Figure 3: Comparison of model outputs on a factual question. Despite retrieving relevant documents with correct information (highlighted in green), RAG is misled by contextual distractors and produces an incorrect answer. MLP Memory generates the correct answer without explicit retrieval.

2020), this memory learns to map the LLM hidden state at a certain step to a vocabulary distribution matching the output of the $k$NN retriever. During inference, the LLM's native output is interpolated with the retriever-pretrained output from the external memory. Our resulting architecture, illustrated in Figure 4, consists of a transformer decoder and an external MLP memory, each pretrained separately with different pretraining tasks. For our pretrained external memory, we aim to achieve the following features simultaneously:

1) **End-to-end differentiability.** Unlike the non-parametric nature of retrievers, our MLP memory is fully parameterized and allows gradieat flow during training. This enables end-to-end joint optimization of the entire model architecture.

2) **Highly compressible memory.** The MLP memory compresses large datastores (e.g., 40TB for 5B tokens in $k$NN-LM) into a compact parametric form (e.g., 4GB for 1B parameters storing 5B tokens), facilitating efficient deployment without performance degradation.

3) **Low inference-time latency.** MLP memory eliminates costly retrieval operations, achieving $2.5\times$ faster inference than RAG methods and $5.6\times$ faster inference than kNN-LM when using a 5B-token retrieval corpus. Crucially, unlike retrieval-based approaches, our method's inference speed remains constant regardless of the retrieval corpus size.

4) **Long-term memory, covering the whole pretraining corpus.** While existing memory tokens serve primarily as working memory by storing local context for immediate use, our MLP memory functions as a long-term repository of generalizable knowledge acquired during the pretraining phase.

Experimental results demonstrate that MLP Memory significantly outperforms existing approaches across multiple dimensions. It achieves average relative improvements of 12.3% (Mistral-7B) and 7.8% (Llama2-7B) on five QA benchmarks, with WebQA showing exceptional gains (37.45% vs. 29.28% baseline). On nine general NLP tasks, it delivers a 5.2 points absolute improvement. MLP Memory also substantially reduces hallucinations on HaluEval, with accuracy improvements of 9.68, 10.08, and 2.14 points on dialogue, QA, and summarization tasks respectively. Most notably, it achieves $2.5\times$ faster time-to-first-token than RAG and $5.6\times$ faster than $k$NN-LM, while maintaining constant inference speed regardless of corpus size, unlike retrieval methods whose latency scales with data size. Figure 1 illustrates MLP Memory's performance gains and inference efficiency over baselines and Figure 3 demonstrates a case where MLP Memory correctly answers factual ques-

tions while RAG fails despite retrieving correct information. These results confirm that parametric compression of retrieval patterns offers a more efficient and effective alternative to explicit retrieval.

## 2 PRELIMINARY: $k$-NEAREST NEIGHBORS LANGUAGE MODEL

The $k$NN-LM (Khandelwal et al., 2020) augments a pre-trained LM by interpolating its parametric distribution with a non-parametric distribution from nearest neighbor retrieval. Given context $c_t = (w_1, ..., w_{t-1})$, and $w_t$ denotes the next token. The next-token probability is:

$$p(w_t \mid c_t) = \lambda \, p_{kNN}(w_t \mid c_t) + (1 - \lambda) \, p_{LM}(w_t \mid c_t), \tag{1}$$

where $\lambda \in [0, 1]$ is the interpolation parameter, $p_{LM}$ is the LM's distribution, and $p_{kNN}$ is retrieval-based distribution.

**Datastore** Constructed via a forward pass over a corpus, the datastore consists of key-value pairs $(k_t, v_t)$ where $k_t = f(c_t)$ encodes context $c_t$ using LM representations, and $v_t$ is the next token $w_t$:

$$(\mathcal{K}, \mathcal{V}) = \{(f(c_t), w_t) \mid (c_t, w_t) \in \mathcal{D}\}. \tag{2}$$

**Inference** The LM encodes context $c$ into query $f(c)$ and retrieves $k$-nearest neighbors $\mathcal{N}$ from $(\mathcal{K}, \mathcal{V})$ using distance metric $d(\cdot, \cdot)$ (typically squared $L^2$). The non-parametric distribution is:

$$p_{kNN}(y \mid c) \propto \sum_{(k_i, v_i) \in \mathcal{N}} \mathbb{I}_{y=v_i} \exp(-d(k_i, f(c))). \tag{3}$$

While $k$NN-LM improves predictions through explicit memory, it suffers from substantial storage requirements and high-latency retrieval. For instance, the Wikitext-103 datastore requires nearly 500 GB of storage even for the GPT2-small model (He et al., 2021). These limitations motivate our MLP Memory, a compact parametric model pretrained to approximate the retrieval function: given a query embedding, it directly outputs a $k$NN-like next token distribution, thereby eliminating both the substantial storage requirements and high-latency retrieval.

## 3 MLP MEMORY

In this section, we present MLP Memory, a lightweight parametric module that learns to internalize retrieval patterns without explicit document access. Our approach consists of three key components: a stack of MLPs that processes hidden representations without token-mixing operations (Section 3.1), a specialized pre-training procedure that enables the MLP to mimic non-parametric retrieval distributions (Section 3.2), and an efficient inference mechanism for deployment (Section 3.3). As illustrated in Figure 4, MLP Memory first learns to mimic non-parametric retrieval distributions during pre-training (Figure 4(b)), then seamlessly integrates with the language model during inference (Figure 4(a)), eliminating both the storage requirements of large datastores and the computational cost of nearest neighbor search.

### 3.1 ARCHITECTURE

Our MLP Memory learns to mimic non-parametric retrieval by mapping query embeddings to $k$NN distributions. Given query $q = f(c)$ from context $c$, the MLP directly predicts $p_{kNN}(y|c)$ without neighbor search, transforming discrete retrieval into a differentiable mapping $\mathcal{M} : \mathbb{R}^d \to \mathbb{R}^{|V|}$, where $d$ is the embedding dimension and $|V|$ is the vocabulary size.

In designing the memory module, we observe from Section 2 that the retriever imitation task processes a single-vector representation without requiring token-mixing operations. Recent studies (Geva et al., 2020) have identified that FFN layers function as key-value memories, suggesting that MLPs play a specialized role in knowledge memorization within LLMs. Based on these insights, we propose pretraining an all-MLP memory that effectively functions as a non-parametric retriever, as illustrated in Figure 4.

The MLP Memory takes hidden representations $f(c)$ from the pretrained LM as input and is trained to predict the corresponding $k$NN distribution $p_{kNN}(y|c)$ as its target. Once trained, the MLP's output distribution is interpolated with the LM's parametric distribution during inference, following the same interpolation scheme as $k$NN-LM but without requiring datastore access or neighbor search.

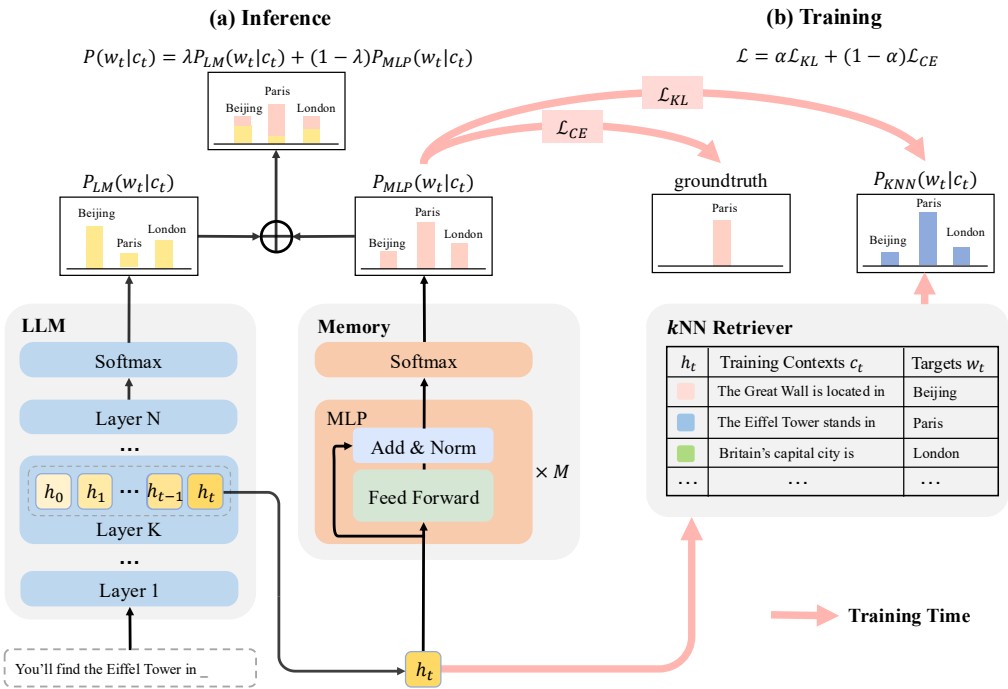

Figure 4: Overview of MLP Memory architecture. (a) Inference: MLP Memory processes context representations from a specific LLM layer, generating token probabilities that are interpolated with LLM outputs for final predictions. (b) Training: MLP Memory learns to imitate retriever behavior using LLM representations as input and distributions generated by $k$NN retrievers as targets, optimized through a hybrid objective.

## 3.2 TRAINING

The training procedure for MLP Memory consists of two primary stages: constructing supervision signals from non-parametric retrieval distributions, and optimizing the MLP to mimic these distributions through a carefully designed loss function.

**Data Construction** To generate supervision for training MLP Memory, we leverage the datastore construction process described in Section 2. We build the datastore $(\mathcal{K}, \mathcal{V})$ through a forward pass over the training corpus, storing context representations and their corresponding next tokens. For each training example $(c_t, w_t) \in \mathcal{D}$, we compute the non-parametric distribution $p_{kNN}(y|c_t)$ by retrieving $k$-nearest neighbors from the datastore. To prevent trivial self-retrieval that would contaminate the learning signal, we exclude the query itself from the neighbor set when constructing the target distribution. These embedding-distribution pairs $\{(f(c_t), p_{kNN}(\cdot|c_t))\}$ are precomputed offline and cached for efficient training.

**Loss Function** Unlike traditional language modeling with single-label targets, $k$NN distributions capture the diversity of plausible continuations by encoding multiple valid next tokens weighted by their contextual similarity. Our ablation studies in Section 5.4 demonstrate that a hybrid objective combining two complementary losses yields optimal performance. Our approach centers on minimizing the Kullback-Leibler divergence (Van Erven & Harremos, 2014) between MLP Memory's output distribution and the cached $k$NN distributions:

$$\mathcal{L}_{KL}(c_t) = \text{KL}(p_{kNN}(\cdot|c_t) \parallel p_{MLP}(\cdot|c_t)) \quad (4)$$

This encourages the memory module to match the full probability distribution rather than merely predicting the most likely token. To prevent excessive deviation from the underlying corpus distribution, we integrate a complementary Cross-Entropy loss (Zhang & Sabuncu, 2018):

$$\mathcal{L}_{CE}(c_t) = -\log p_{MLP}(w_t|c_t) \quad (5)$$

The final training objective balances these two components through a hyperparameter $\alpha$:

$$\mathcal{L}(c_t) = \alpha \cdot \mathcal{L}_{KL}(c_t) + (1 - \alpha) \cdot \mathcal{L}_{CE}(c_t) \tag{6}$$

The KL term encourages learning distributional patterns while the CE term ensures accurate ground-truth prediction, preventing the overfitting that occurs with cross-entropy alone.

## 3.3 INFERENCE

Once trained, MLP Memory integrates with the base language model through simple probability interpolation. During inference, MLP Memory processes hidden representations from the language model $\mathcal{M}_{\text{LM}}$ and produces a distribution that is interpolated with the LM's output:

$$p_{final}(w_t|c_t) = \lambda \cdot p_{MLP}(w_t|c_t) + (1 - \lambda) \cdot p_{LM}(w_t|c_t) \tag{7}$$

where $\lambda \in [0, 1]$ controls the influence of retrieval-based knowledge.

Unlike retrieval-augmented approaches that require nearest neighbor search and extended context processing, MLP Memory requires only a single forward pass through a lightweight all-MLP architecture. As demonstrated in Figure 1, our method achieves $2.5\times$ faster time-to-first-token than RAG (top-5) and $5.6\times$ faster than $k$NN-LM, despite $k$NN-LM employing dimension reduction from 4096 to 256 for acceleration. For tokens per second, MLP Memory delivers $1.5\times$ higher throughput than RAG and $6\times$ higher than $k$NN-LM, while introducing only $1.2\times$ overhead relative to the base model. Crucially, this performance remains constant regardless of retrieval corpus size, unlike retrieval-based methods whose latency scales with datastore size.

## 4 EXPERIMENTAL SETUP

**Overview** We conduct comprehensive experiments to evaluate MLP Memory across four critical dimensions. First, we assess performance on five question-answering benchmarks (5.1) to demonstrate that our approach represents a novel form of parametric memory that surpasses both traditional parametric methods (continued pretraining, LoRA) and non-parametric approaches (RAG). Second, we evaluate on fundamental NLP tasks (5.2) to verify that integrating MLP Memory preserves the base model's general capabilities. Third, we examine hallucination reduction (5.3) on HaluEval to validate our method's effectiveness in improving factual accuracy. Finally, we present an ablation study (5.4) to analyze design choices such as loss weighting and layer selection.

**Implementation Details** We conduct our experiments on 32×A800 80GB GPUs. To demonstrate the generalizability of our approach, we employ two distinct backbone models: Llama-2-7B (Touvron et al., 2023) and Mistral-7B-v0.3 (Jiang et al., 2023). For question-answering benchmarks, we build key-value datastores and non-parametric distributions using both models on preprocessed Wikipedia-2021 (Izacard et al., 2022), and train separate 1B-parameter MLP Memory modules with learning rate 4e-4. The MLP Memory uses 8 layers by default. See Appendix L for details. For general NLP tasks, we build datastores using Mistral-7B-v0.3 on a heterogeneous corpus following Geng et al. (2024), and train the MLP Memory with learning rate 4e-4. For hallucination evaluation, we directly apply the MLP Memory trained from question-answering experiments. All experiments use a training budget equivalent to the computational cost of training a 7B parameter model for 1 epoch. The training hyperparameter $\alpha$ is set to 0.4 across all tasks. The interpolation hyperparameter $\lambda$ is tuned on the validation split of each task following Khandelwal et al. (2020), see more details in Appendix D.

**Baselines** We compare MLP Memory against established methods for improving factual accuracy and knowledge utilization: **RAG**, which employs BGE (Chen et al., 2024) as the retrieval model and retrieves top-5 documents to ensure comprehensive context coverage. $k$**NN-LM** (Khandelwal et al., 2020), configured with interpolation parameter $\lambda = 0.1$ and temperature $\tau = 10.0$ following (Geng et al., 2024). **LoRA** (Hu et al., 2022), applied to query, key, value, and MLP layers, with rank adjusted to match the parameter count of our MLP Memory modules. **Continued Pretraining (CPT)**, which involves further training of all model parameters on the corresponding corpus.

Table 1: Question answering performance across five benchmarks. Positive gains are shown in green and negative changes in red. Percentage in parentheses denotes the relative improvement over the base model. All methods use the same Wikipedia-2021 corpus for training or retrieval.

| Methods | Open-Domain QA | | | Long-form QA | Multihop QA | Average |
|---|---|---|---|---|---|---|
| | NQ | WebQA | TriviaQA | TruthfulQA | HotpotQA | |
| Llama2-7B | 23.18 | 32.09 | 56.91 | 29.16 | 22.72 | 32.81 |
| *Non-parametric methods* | | | | | | |
| +RAG | $14.60_{-8.58}$ | $36.71_{+4.62}$ | $62.20_{+5.29}$ | $31.59_{+2.43}$ | $19.60_{-3.12}$ | $32.94(+0.4\%)$ |
| +$k$NN-LM | $23.16_{-0.02}$ | $33.46_{+1.37}$ | $57.31_{+0.40}$ | $29.22_{+0.06}$ | $22.66_{-0.06}$ | $33.16(+1.1\%)$ |
| *Parametric methods* | | | | | | |
| +CPT | $12.90_{-10.28}$ | $31.55_{-0.54}$ | $58.81_{+1.90}$ | $29.56_{+0.40}$ | $15.49_{-7.23}$ | $29.66(-9.6\%)$ |
| +LoRA | $17.88_{-5.30}$ | $35.19_{+3.10}$ | $58.14_{+1.23}$ | $28.33_{-0.83}$ | $17.18_{-5.54}$ | $31.34(-4.5\%)$ |
| +MLP Mem | $27.04_{+3.86}$ | $36.61_{+4.52}$ | $57.50_{+0.59}$ | $30.04_{+0.88}$ | $25.69_{+2.97}$ | $\mathbf{35.38}(+7.8\%)$ |
| Mistral-7B-v0.3 | 20.63 | 29.28 | 57.65 | 32.09 | 20.96 | 32.12 |
| *Non-parametric methods* | | | | | | |
| +RAG | $22.56_{+1.93}$ | $24.90_{-4.38}$ | $54.21_{-3.44}$ | $35.47_{+3.38}$ | $29.77_{+8.81}$ | $33.38(+3.9\%)$ |
| +$k$NN-LM | $21.05_{+0.42}$ | $30.51_{+1.23}$ | $57.77_{+0.12}$ | $32.33_{+0.24}$ | $21.20_{+0.24}$ | $32.57(+1.4\%)$ |
| *Parametric methods* | | | | | | |
| +CPT | $12.16_{-8.47}$ | $34.06_{+4.78}$ | $61.21_{+3.56}$ | $29.18_{-2.91}$ | $16.04_{-4.92}$ | $30.53(-5.0\%)$ |
| +LoRA | $18.17_{-2.46}$ | $34.50_{+5.22}$ | $61.60_{+3.95}$ | $30.91_{-1.18}$ | $16.23_{-4.73}$ | $32.28(+0.5\%)$ |
| +MLP Mem | $25.20_{+4.57}$ | $37.45_{+8.17}$ | $60.99_{+3.34}$ | $32.54_{+0.45}$ | $24.14_{+3.18}$ | $\mathbf{36.06}(+12.3\%)$ |

Table 2: Performance on nine general NLP tasks spanning sentiment classification, textual entailment, and topic classification. ↑ indicate improvement over the Mistral-7B-v0.3 baseline, while ↓ indicate decreased performance.

| Methods | Sentiment Classification | | | | | Textual. | | Topic. | | Average |
|---|---|---|---|---|---|---|---|---|---|---|
| | SST2 | MR | CR | RT | HYP | CB | RTE | AGN | Yahoo | |
| Mistral-7B-v0.3 | 81.21 | 75.35 | 62.30 | 74.95 | 55.42 | 69.64 | 59.57 | 75.95 | 56.36 | 67.86 |
| *Non-parametric methods* | | | | | | | | | | |
| +RAG | 87.20↑ | 83.70↑ | 71.55↑ | 82.36↑ | 54.65↓ | 57.14↓ | 66.43↑ | 75.64↓ | 58.43↑ | 70.79↑ |
| +$k$NN-LM | 82.15↑ | 76.85↑ | 61.70↓ | 74.95 | 56.78↑ | 71.42↑ | 60.28↑ | 76.13↑ | 56.26↓ | 68.50↑ |
| *Parametric methods* | | | | | | | | | | |
| +CPT | 87.09↑ | 82.85↑ | 82.60↑ | 77.48↑ | 60.65↑ | 57.14↓ | 52.71↓ | 83.10↑ | 51.56↓ | 70.58↑ |
| +LoRA | 86.54↑ | 83.20↑ | 75.10↑ | 79.83↑ | 55.42 | 51.78↓ | 56.31↓ | 65.46↓ | 57.30↑ | 67.88↑ |
| +MLP Mem | 83.19↑ | 79.90↑ | 75.95↑ | 75.42↑ | 64.15↑ | 76.79↑ | 64.62↑ | 80.28↑ | 57.33↑ | $\mathbf{73.07}$↑ |

## 5 EXPERIMENTAL RESULTS

### 5.1 QUESTION ANSWERING PERFORMANCE

We evaluate MLP Memory on five diverse QA benchmarks: Natural Questions (NQ) (Kwiatkowski et al., 2019), WebQA (Berant et al., 2013), TriviaQA (Joshi et al., 2017), TruthfulQA (Lin et al., 2022), and HotpotQA (Yang et al., 2018), comparing against CPT, LoRA, and RAG. As shown in Table 1, Mistral-7B-v0.3 with MLP Memory achieves an average relative improvement of 12.3% over the baseline across five benchmarks, with particularly striking improvements on NQ (25.20% vs. baseline 20.63%) and WebQA (37.45% vs. baseline 29.28%). While CPT and LoRA suffer significant degradation across all tasks—likely due to catastrophic forgetting during domain-specific training—MLP Memory maintains or improves performance by learning to emulate retrieval behavior without modifying the base model's parameters. Notably, our approach outperforms both RAG and $k$NN-LM, even though they leverage the same Wikipedia-2021 corpus for retrieval at inference time, suggesting that our parametric compression of retrieval patterns captures richer contextual relationships than explicit document retrieval. The consistent gains across both factoid QA (NQ, TriviaQA) and multi-hop reasoning (HotpotQA) demonstrate that MLP Memory effectively bridges the gap between parametric and non-parametric memory systems.

Table 3: Performance on HaluEval benchmark across question answering, dialogue, and summarization tasks. Results show accuracy (%). RAG is not evaluated on summarization as this task requires only the source document.

|                | Dialogue | QA | Summarization |
|----------------|----------|-----|---------------|
| Mistral-7B-v0.3 | 57.18 | 53.99 | 50.27 |
| +CPT | 51.68$_{-5.50}$ | 46.49$_{-7.50}$ | 47.39$_{-2.88}$ |
| +LoRA | 55.51$_{-1.67}$ | 50.02$_{-3.97}$ | 50.38$_{+0.11}$ |
| +RAG | 59.06$_{+1.88}$ | 65.09$_{+11.10}$ | - |
| +MLP Mem | 66.86$_{+9.68}$ | 64.07$_{+10.08}$ | 52.41$_{+2.14}$ |

## 5.2 GENERAL NLP TASK PERFORMANCE

To ensure MLP Memory doesn't compromise fundamental language understanding, we evaluate on nine standard NLP tasks spanning sentiment classification (SST-2 (Socher et al., 2013), MR (Pang & Lee, 2005a), CR (Hu & Liu, 2004), RT (Pang & Lee, 2005b), HYP (Kiesel et al., 2019)), textual entailment (CB (De Marneffe et al., 2019), RTE (Dagan et al., 2010)), and topic classification (AGNews (Zhang et al., 2015a), Yahoo (Zhang et al., 2015b)). Table 2 reveals that MLP Memory achieves comprehensive improvements across all general tasks, achieving the highest average score compared to all baselines. The improvements are particularly pronounced on reasoning-intensive tasks like RTE (64.62% vs. baseline 59.57%) and CB (76.79% vs. baseline 69.64%), suggesting that the retrieval patterns learned by MLP Memory provide useful inductive biases even for tasks that don't explicitly require factual knowledge retrieval. In contrast, CPT and LoRA show mixed results with improvements on some tasks but degradation on others. The robust performance across this diverse task suite demonstrates that MLP Memory's external parametric memory complements rather than interferes with the base model's learned representations.

## 5.3 HALLUCINATION REDUCTION

We assess MLP Memory's ability to reduce hallucinations using HaluEval (Li et al., 2023) across three generation tasks: dialogue, question answering, and summarization, where models must identify factual inconsistencies in generated content. As shown in Table 3, parametric methods (CPT and LoRA) severely degrade performance, confirming the risks of weight modification. MLP Memory consistently improves hallucination detection across all three domains, with gains of 9.68, 10.08, and 2.14 points respectively. These substantial improvements indicate that the retrieval patterns encoded in MLP Memory significantly help the model better distinguish factual from hallucinated content. The effectiveness across diverse generation contexts—from free-form dialogue to constrained summarization—suggests that MLP Memory's learned retrieval behavior provides a general mechanism for grounding language generation in factual knowledge. This hallucination reduction, combined with strong QA performance and enhanced general capabilities, validates our core hypothesis that decoupling memorization from reasoning through retriever-pretrained external memory can enhance factual accuracy without the typical trade-offs of parametric or non-parametric approaches.

## 5.4 ABLATION STUDY

**Ablation Setup** We conduct ablation experiments across three GPT2 (Radford et al., 2019) scales: small (12 layers), medium (24 layers), and large (36 layers), paired with corresponding MLP Memory modules of 117M, 345M, and 774M parameters respectively. All experiments are evaluated on WikiText-103 (Merity et al., 2016) to investigate loss weighting and optimal layer selection.

**Impact of Loss Weighting** We examine how balancing KL and CE losses affects retriever imitation by varying $\alpha$ from 0.0 to 1.0. As Figure 5(a) shows, extreme values produce suboptimal results—low values prevent the MLP memory from learning from the $k$NN distribution, while high values cause overfitting to the language modeling objective. The optimal balance occurs at $\alpha = 0.4$, indicating both objectives are necessary. KL divergence leverages the information-rich kNN distribution, enabling more effective generalization, while CE loss provides essential token-level prediction accuracy. This balanced approach prevents overfitting while maintaining predictive power.

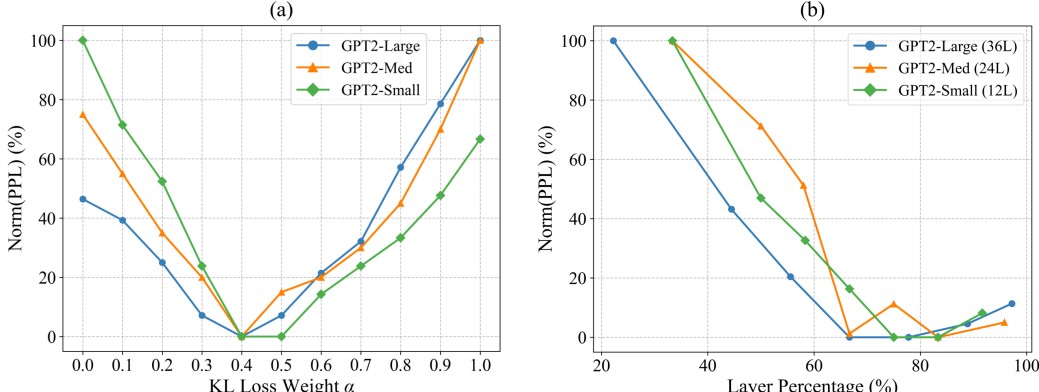

Figure 5: (a) Impact of KL loss weight $\alpha$ on retriever imitation. Lower PPL (min-max normalized for clarity) indicates better performance, with optimal balance at $\alpha = 0.4$. (b) Impact of input layer depth on MLP Memory performance across model sizes. Layer percentage denotes depth in the decoder stack (e.g., 70% corresponds to layer 25 in GPT2-large).

**Which Layer Provides the Best Representation for MLP Memory?**   While $k$NN-LM performs best using the input to the final feedforward layer as the retrieval key, our MLP Memory consistently achieves optimal performance at around 70% of network depth, regardless of model scale. Our finding aligns with Memorizing Transformers (Burtsev et al., 2021), which also selected around 75% depth for optimal retrieval performance. We evaluate GPT2-small (12 layers), GPT2-medium (24 layers), and GPT2-large (36 layers), attaching the MLP Memory to various transformer blocks. As shown in Figure 5(b), the x-axis indicates relative depth (20%–100%), and the y-axis shows min-max normalized perplexity (0% = best, 100% = worst). This consistent pattern across all model sizes contrasts with the $k$NN-LM convention of using final-layer representations.

## 6   RELATED WORK

**Retrieval-Augmented Generation**   RAG (Lewis et al., 2021; Peng et al., 2023; Gao et al., 2022) mitigates hallucinations by grounding generation in external knowledge. Despite improving factual accuracy, RAG faces limitations: retrieval latency, coarse granularity, and limited LLM integration (Zhang et al., 2024). Recent work (Su et al., 2025) explores enhanced retrieval with LLM priors. Our approach proposes a parametric memory mimicking non-parametric retrieval, eliminating explicit document retrieval while preserving knowledge augmentation.

**Memory-Augmented Language Models**   Various architectures explored memory augmentation, from Memory Networks (Weston et al., 2015) with explicit read-write components to Memory Transformers (Burtsev et al., 2021) with extended attention. LongMem (Wang et al., 2023) and MemoRAG (Qian et al., 2025) introduced decoupled architectures for long-term history storage. While these focus on context extension, our MLP memory expands across the entire pre-training corpus, enabling long-term generalizable knowledge storage.

**MLP Architectures**   All-MLP architectures emerged as transformer alternatives, with gMLP (Liu et al., 2021) matching transformer performance and sparse MLPs (Yu et al., 2022) showing superior training efficiency. Studies (Geva et al., 2020) identified FFN layers as key-value memories in LLMs. Inspired by this, we propose pretraining an all-MLP memory as a non-parametric retriever, leveraging MLPs' memorization capabilities for a compact, differentiable knowledge store.

## 7   CONCLUSION

In this paper, we introduced MLP Memory, a novel approach for enhancing language models by learning to internalize retrieval patterns. By pretraining a lightweight MLP module to imitate kNN retriever behavior on the entire pretraining corpus, MLP Memory captures the benefits of retrieval-augmented generation in a fully parametric form, without requiring explicit document access.

The key advantage of MLP Memory lies in its efficiency and effectiveness. Our approach achieves 12.3% relative improvement on question-answering benchmarks, 5.2 points gain on general NLP tasks, and up to 10 points reduction in hallucinations—while delivering $2.5\times$ faster inference than RAG and maintaining constant speed regardless of corpus size. Unlike parametric fine-tuning that risks catastrophic forgetting or non-parametric RAG that suffers from high latency, MLP Memory enhances model capabilities without these typical trade-offs.

MLP Memory introduces a new paradigm for language model enhancement that fundamentally reimagines how models access and utilize knowledge. By parametrically encoding retrieval behavior through a pretrained memory component, our approach creates a more efficient, accurate, and scalable framework that bridges the gap between parametric and non-parametric methods.

## ACKNOWLEDGMENTS

This work is sponsored by the National Natural Science Foundation of China (NSFC) grant (No. 62576211) and the National Key Research and Development Program of China (No. 2023ZD0121402). It is also the result of a collaborative project on novel language model architectures between Shanghai Jiao Tong University (SJTU) and the Shanghai Artificial Intelligence Laboratory. The computational resources required for pretraining the models were provided by the Shanghai AI Lab. This work is also supported by the Specialized Program on Fundamental Research from Science and Technology Commission of Shanghai Municipality (No. 2025SHZDZX025G09).

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

## A  THE USE OF LARGE LANGUAGE MODELS

We utilized ChatGPT as an editorial assistant to enhance the manuscript's language quality, correct grammatical errors, and ensure academic writing standards. All scientific contributions, including the research design, experimental procedures, and analytical interpretations, were originated and performed independently by the authors.

## B  REPRODUCIBILITY STATEMENTS

We provide comprehensive implementation details of our method throughout the paper. Our method is thoroughly described in Section 3. We present detailed settings for our experiments and analyses in Section 4,  5.4 and Appendix C. Code and checkpoints will be released upon acceptance.

## C  IMPLEMENTATION DETAILS

**Datasets**    For the general NLP tasks in Section 5.2, we utilize a heterogeneous corpus constructed by aggregating several publicly available sources that cover diverse domains relevant to common NLP tasks. Following the methodology from (Geng et al., 2024), this corpus comprises WikiText-103 Merity et al. (2016) for encyclopedic content, Amazon Reviews He & McAuley (2016) for user-generated product feedback, CC-NEWS Hamborg et al. (2017) for journalistic content, and IMDB Maas et al. (2011) for movie reviews and discussions.

This diverse mixture captures both formal and informal language patterns, spans multiple domains from news articles to consumer opinions, and provides comprehensive coverage of linguistic phenomena encountered in real-world NLP applications. The complete dataset is publicly available at: `https://huggingface.co/datasets/wentingzhao/knn-prompt-datastore`.

**Evaluation Metrics**    For question answering benchmarks, following  Cheng et al. (2024), we evaluate three Open Domain Question Answering datasets and HotpotQA using the Exact Match (EM) metric. For long-form QA evaluation, we employ three complementary metrics: MC1, which measures whether the model assigns the highest likelihood to the most accurate answer; MC2, which sums the normalized probabilities over all correct answers; and MC3, which evaluates whether the model assigns a higher average likelihood to true answers than to false ones. We report the average of these three metrics as the final performance measure for long-form QA tasks. For general NLP tasks, following the methodology from  Shi et al. (2022b), we report results using the domain-conditional PMI scoring rule Holtzman et al. (2021). For hallucination reduction evaluation, we use accuracy as the primary metric to assess the model's ability to generate factually correct responses.

**Hyperparameters**    In Table 4, we list the hyperparameters used for training the 1B MLP Memory model (excluding embedding parameters).

Table 4: Hyperparameters for training the 1B MLP Memory model.

| Hyperparameter | Assignment |
|---|---|
| optimizer | AdamW |
| learning rate | 4e-4 |
| lr scheduler type | linear |
| warmup ratio | 0.03 |
| weight decay | 0.0 |
| epochs | 5 |
| flash attention | False |
| batch size | 4 |
| gradient accumulation steps | 4 |
| num GPUs | 32 |
| max train samples | 33,000,000 |

## D    SENSITIVITY TO INTERPOLATION WEIGHT $\lambda$

We conducted a comprehensive analysis of our method's sensitivity to the interpolation weight $\lambda$ on the HaluEval benchmark using Mistral-7B-v0.3. Table 5 presents the results across three tasks: Dialogue, QA, and Summarization, with $\lambda$ values ranging from 0.1 to 0.9.

Our findings demonstrate that the method exhibits robust performance across a wide range of $\lambda$ values, with optimal performance generally observed in the 0.35-0.55 range. Specifically, the Dialogue task achieves its best performance at $\lambda = 0.45$ (64.07%), QA peaks at $\lambda = 0.55$ (66.86%), and Summarization reaches its maximum at $\lambda = 0.35$ (52.41%). Notably, all three tasks show consistent improvements over the baseline Mistral-7B-v0.3 model across the optimal range, with QA showing the most substantial gains (up to 10.08 points improvement).

The performance remains relatively stable within the 0.3-0.6 range, with only gradual degradation outside this interval. When $\lambda$ approaches extreme values (e.g., 0.9), performance deteriorates significantly, particularly for Dialogue and Summarization tasks, though still maintaining improvements over the baseline in the QA task.

These results confirm that our method is not overly sensitive to the specific choice of $\lambda$ within a reasonable range, making it practical for deployment without extensive hyperparameter tuning. The consistent improvements across different $\lambda$ values and tasks validate the robustness of our approach.

Table 5: Performance sensitivity analysis of interpolation weight $\lambda$ on HaluEval benchmark using Mistral-7B-v0.3. Results are reported as accuracy (%) across three tasks: Dialogue, QA, and Summarization. The first row shows baseline Mistral-7B-v0.3 performance without memory augmentation. Bold values indicate the best performance for each task.

| $\lambda$ | Dialogue | QA | Summarization |
|---|---|---|---|
| Mistral-7B-v0.3 | 57.18 | 53.99 | 50.27 |
| 0.10 | 56.80 | 59.86 | 50.92 |
| 0.20 | 59.43 | 62.01 | 51.87 |
| 0.30 | 61.99 | 64.11 | 52.17 |
| 0.35 | 63.01 | 64.96 | **52.41** |
| 0.40 | 63.88 | 66.03 | 52.11 |
| 0.45 | **64.07** | 66.55 | 51.55 |
| 0.50 | 63.57 | 66.57 | 51.39 |
| 0.55 | 63.24 | **66.86** | 50.73 |
| 0.60 | 62.38 | 66.30 | 49.79 |
| 0.70 | 59.65 | 64.77 | 47.72 |
| 0.80 | 56.42 | 62.78 | 46.53 |
| 0.90 | 49.71 | 60.36 | 46.67 |

## E    INFERENCE EFFICIENCY ANALYSIS

Table 6 presents the computational cost breakdown for both Transformer and MLP architectures in terms of FLOPs per token. As demonstrated, the primary difference in computational efficiency stems from the absence of attention mechanisms in pure MLP models.

By comparing these computational requirements, we derive the theoretical speed ratio between the Transformer (denoted as $FLOPs_t$) and the MLP models (denoted as $FLOPs_m$):

$$\frac{FLOPs_t}{FLOPs_m} \approx \frac{4n_{layer}d_{model}(2d_{attn} + d_{ff}) + 2n_{layer}n_{ctx}d_{attn}}{6n_{layer}d_{model}d_{ff}} = 1 + \frac{n_{ctx}}{12d_{model}}, \quad (8)$$
$$\text{with the standard} \quad d_{attn} = d_{ff}/4 = d_{model}.$$

This relationship in Equation 8 reveals that MLPs maintain a consistent computational advantage across all context lengths, with the efficiency gap widening as context length increases.

Table 6: Flops per Token at inference time. Following Kaplan et al. (2020), we analyze computational requirements for Transformer and MLP architectures where $n_{layer}$(number of layers), $d_{model}$(dimension of the residual stream), $d_{ff}$(dimension of the intermediate feed-forward layer), $d_{attn}$(dimension of the attention output) , $n_{heads}$(number of attention heads per layer), $n_{ctx}$(the length of input context), $n_{vocab}$(vocabulary size). $C_{forward}$ denotes computational cost per token inference step.

| Openration | FLOPs per Token(Transformer) | FLOPs per Token(MLP) |
|---|---|---|
| Embed | $4d_{model}$ | $-$ |
| Attention: QKV | $2n_{layer}d_{model}3d_{attn}$ | $-$ |
| Attention: Mask | $2n_{layer}n_{ctx}d_{attn}$ | $-$ |
| Attention: Project | $2n_{layer}d_{attn}d_{model}$ | $-$ |
| Feedforward | $2n_{layer}2d_{model}d_{ff}$ | $3n_{layer}2d_{model}d_{ff}$ |
| De-embed | $2d_{model}n_{vocab}$ | $2d_{model}n_{vocab}$ |
| Total(Non-Embedding) | $C_{forward} = 4n_{layer}d_{model}(2d_{attn} + d_{ff})$ $+2n_{layer}n_{ctx}d_{attn}$ | $C_{forward} = 6n_{layer}d_{model}d_{ff}$ |

## F  SCALING LAW

**Setup**  We conduct scaling law experiments using standard decoder-only models and our overall model architecture. As baselines, we use four GPT-2 Radford et al. (2019) variants with increasing parameter counts: GPT2-small (124M), GPT2-medium (345M), GPT2-large (774M), and GPT2-xl (1.5B). For MLP Memory, we define three configurations: small (124M), medium (335M), and large (774M) that align with the scaling trend of standard architectures. The MLP Memory module is externally integrated with a matching-sized GPT-2 variant, resulting in total parameter counts of approximately 248M, 710M, and 1.5B for the small, medium, and large configurations, respectively. All models are trained on two datasets: WikiText-103 Merity et al. (2016) (around 100M tokens) and a mixed Web dataset (around 600M tokens). Following  Shi et al. (2022a), our Web dataset combines diverse knowledge sources relevant to common NLP tasks, including WikiText-103, Amazon Reviews He & McAuley (2016), CC-NEWS Hamborg et al. (2017), and IMDB Maas et al. (2011).

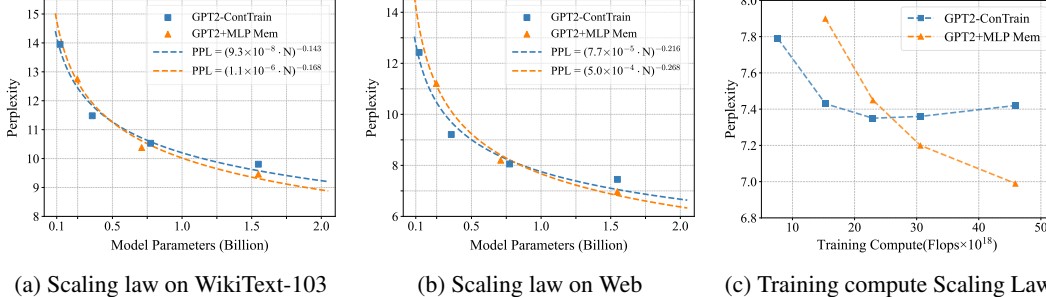

(a) Scaling law on WikiText-103     (b) Scaling law on Web     (c) Training compute Scaling Law

Figure 6: Power-law scaling behavior with model size $N$ and training compute $C$. (a) Scaling results compare the continued training of GPT2 (GPT2-ConTrain) with our overall model architecture (GPT2+MLP Mem) under fixed compute. Our fitted curve shows a 17.5% exponent improvement on WikiText-103. (b) On the larger Web dataset, our architecture exhibits stronger scaling gains from increased data size, with an exponent improvement of 24.1%. (c) At the GPT2-xl scale, our architecture continues to benefit from additional training on the Web dataset without overfitting.

**Scaling law with model parameters** $N$   Following Kaplan et al. (2020), we model perplexity scaling as $PPL = (\beta \cdot N)^{\gamma}$. Under fixed compute, we compare our architecture to continued GPT-2 training on WikiText-103 and Web datasets in terms of test perplexity scaling with model size $N$. Results in Figure6 show our architecture demonstrates a steeper scaling curve than the decoder-only model, indicating improved scaling efficiency. The power-law scaling laws on WikiText-103 can be expressed as:

$$PPL_d = (9.3 \cdot 10^{-8}N)^{-0.143} \quad and \quad PPL_m = (1.1 \cdot 10^{-6}N)^{-0.168} \tag{9}$$

where $PPL_d$ and $PPL_m$ denote the test perplexity of the decoder model and our architecture, respectively. The corresponding power-law scaling laws on the Web dataset are as follows:

$$PPL_d = (7.7 \cdot 10^{-5} N)^{-0.216} \quad and \quad PPL_m = (5.0 \cdot 10^{-4} N)^{-0.268} \tag{10}$$

These results highlight the superior scaling efficiency of our overall model architecture compared to the standard decoder-only baseline, on both WikiText-103 and the Web dataset.

**Scaling law with training compute** $C$  We further examine how model performance scales with training compute $C$ while keeping model size fixed. At the GPT2-xl scale, we conduct experiments on the Web dataset, measuring test perplexity after varying amounts of training flops. As illustrated in Figure 6 (c), our overall model architecture achieves significantly lower perplexity with increasing training compute, with no signs of overfitting. This suggests that the retriever imitation pretraining task is more challenging and continues to benefit from additional compute.

## G  COMPARING OUTPUT DISTRIBUTION CHARACTERISTICS OF LM, $k$NN, AND MLP MEMORY

As two samples illustrated in Figure 7, distributions produced by LM, $k$NN search, and MLP Memory exhibit distinct characteristics. LM typically yields smooth and dense probability distributions, as it is trained to generalize across large corpora and capture broad contextual patterns.

In contrast, the kNN approach produces sparse and spiky distributions, concentrating most of the probability mass on only a few retrieved neighbors. For instance, when using a GPT-2 model (vocabulary size 50,257), even after retrieving the top-$k$ neighbors (e.g., $k = 1024$), only a small subset of these neighbors meaningfully influences the output distribution, while the majority receive near-zero probability.

The MLP Memory lies between LMs and kNN in terms of distribution characteristics. As a neural model, it is trained using a combination of KL loss and CE loss to approximate the sparse and spiky distributions produced by the kNN approach. While its outputs remain somewhat smoother due to the training objective, the resulting distributions are sharper than those of standard LMs, yet not as extreme as the highly concentrated outputs of kNN.

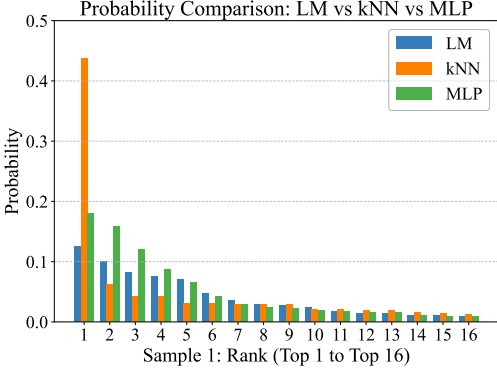 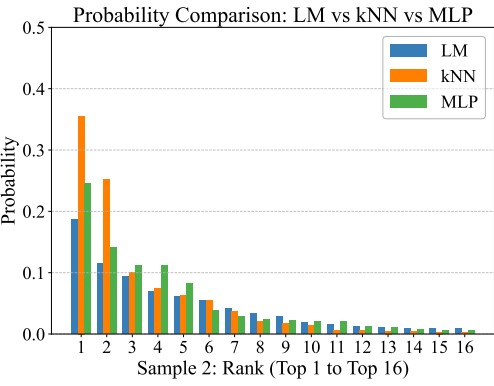

Figure 7: Comparison of output probability distributions. Two samples show the top-16 probabilities from the LM and $k$NN distributions using GPT2-large, along with the distribution generated by the MLP Memory based on the same large model size.

Table 7 compares the output sparsity of LM, $k$NN, and MLP Memory by reporting the number of tokens assigned non-zero probabilities at various thresholds. The LM assigns non-zero mass to all 50,257 tokens, reflecting its dense distribution. However, this number drops sharply at higher thresholds, with only 2 tokens receiving probabilities above 0.1, indicating a rapid decay despite its broad support.

Table 7: Number of tokens with non-zero probability mass at different thresholds. This table reports the number of tokens assigned non-zero probabilities by the LM, $k$NN, and MLP Memory, across a range of probability thresholds. All values are averaged over 20,000 test samples.

| Types | $> 0.0$ | $> 10^{-6}$ | $> 10^{-5}$ | $> 10^{-4}$ | $> 10^{-3}$ | $> 10^{-2}$ | $> 10^{-1}$ |
|-------|---------|-------------|-------------|-------------|-------------|-------------|-------------|
| LM | 50257 | 1760 | 562 | 148 | 34 | 7 | 2 |
| $k$NN | 251 | 217 | 197 | 146 | 43 | 9 | 2 |
| MLP | 50257 | 1151 | 388 | 115 | 30 | 7 | 2 |

In contrast, the $k$NN output is highly sparse, with only 251 tokens assigned any non-zero probability. Even at low thresholds (e.g., $10^{-6}$), the number remains limited, confirming its concentrated nature shaped by a small set of retrieved neighbors.

MLP Memory exhibits intermediate behavior. Although it outputs over the full vocabulary like the LM, the number of tokens exceeding higher thresholds aligns more closely with $k$NN. This suggests that MLP Memory learns to approximate the spiky distributions of $k$NN while maintaining some smoothness from its parametric formulation.

Table 8: Cumulative token count required to reach probability mass thresholds. This table indicates the number of top-ranked tokens needed to accumulate a total probability mass exceeding thresholds such as 0.8, 0.9, etc. All values are averaged over 20,000 test samples.

| Types | Top Prob Count(sum $> 0.8$) | sum $> 0.9$ | sum $> 0.95$ | sum $> 0.99$ |
|-------|------------------------------|-------------|--------------|--------------|
| LM | 23 | 63 | 142 | 617 |
| $k$NN | 22 | 43 | 68 | 126 |
| MLP | 13 | 33 | 72 | 308 |

Table 8 further examines distribution sharpness by reporting the number of top-ranked tokens needed to accumulate a specified proportion of total probability mass. Here, we observe that the $k$NN distribution reaches 99% cumulative probability with only 126 tokens, while the language model (LM) requires 617 tokens to achieve the same threshold. This suggests that the LM's probability mass is more broadly spread across the vocabulary, in contrast to the highly concentrated outputs of $k$NN.

Interestingly, the MLP Memory achieves 99% cumulative probability with 308 tokens, placing it between LM and $k$NN. Notably, the MLP reaches 80% total probability with only 13 tokens—fewer than both LM and $k$NN—indicating that it captures prominent signals more efficiently. These results support the observation that MLP Memory produces sharper distributions than LMs, yet avoids the extreme sparsity of $k$NN.

## H  EFFECT OF DIFFERENT $k$NN TARGET DISTRIBUTIONS

Figure 8 presents the test perplexity of our overall model architecture evaluated at various training steps. In all settings, both the base language model and the MLP Memory are of small size (GPT2-small), with the MLP Memory trained to mimic $k$NN target distributions constructed from different base models: GPT2-small, GPT2-medium, GPT2-large, and GPT2-xl. As training progresses, test perplexity steadily declines across all variants, indicating stable optimization and effective learning. Among them, the model trained on the $k$NN-XL distribution achieves the lowest final test perplexity (12.84), closely followed by the one trained on $k$NN-large (12.85). In contrast, the models trained on $k$NN-medium and $k$NN-small converge to higher perplexities of approximately 12.87 and 12.91, respectively.

These results demonstrate that $k$NN target distributions derived from larger base models lead to improved performance when used to train the MLP Memory. The richer and more informative supervision encoded in these distributions appears to enhance the parametric memory's generalization ability.

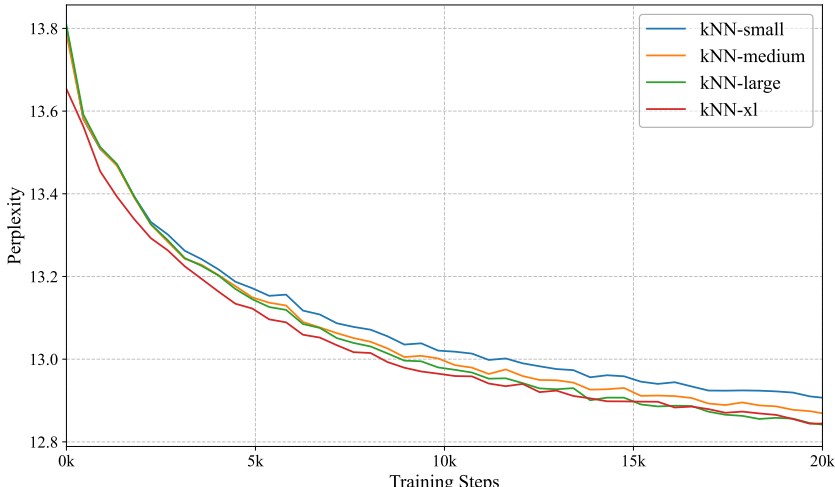

Figure 8: Test perplexity of our overall model architecture, where both the base language model and the MLP Memory are of small size (GPT2-small). The MLP Memory is trained to mimic different $k$NN target distributions constructed from various base models: $k$NN-small (GPT2-small), $k$NN-med (GPT2-medium), $k$NN-large (GPT2-large), and $k$NN-XL (GPT2-xl).

## I  SENSITIVITY TO $k$ IN TARGET DISTRIBUTION GENERATION

We used $k = 1024$ for generating all target distributions. Table 9 shows the sensitivity analysis using GPT2-large-CPT, namely GPT2-large with continued pre-training on WikiText-103. While smaller $k$ values degrade performance, values beyond 1024 yield minimal gains while significantly increasing computational costs, making $k = 1024$ optimal for practical deployment.

Table 9: Test perplexity sensitivity to different values of $k$ in target distribution generation using GPT2-large-CPT on WikiText-103.

| Models | $k$ | Perplexity |
|---|---|---|
| GPT2-large-CPT | – | 10.43 |
| +$k$NN-LM | 1 | 10.30 |
| | 2 | 10.11 |
| | 4 | 9.95 |
| | 8 | 9.83 |
| | 16 | 9.71 |
| | 32 | 9.63 |
| | 64 | 9.57 |
| | 128 | 9.52 |
| | 256 | 9.48 |
| | 512 | 9.46 |
| | 1024 | 9.43 |
| | 2048 | 9.42 |

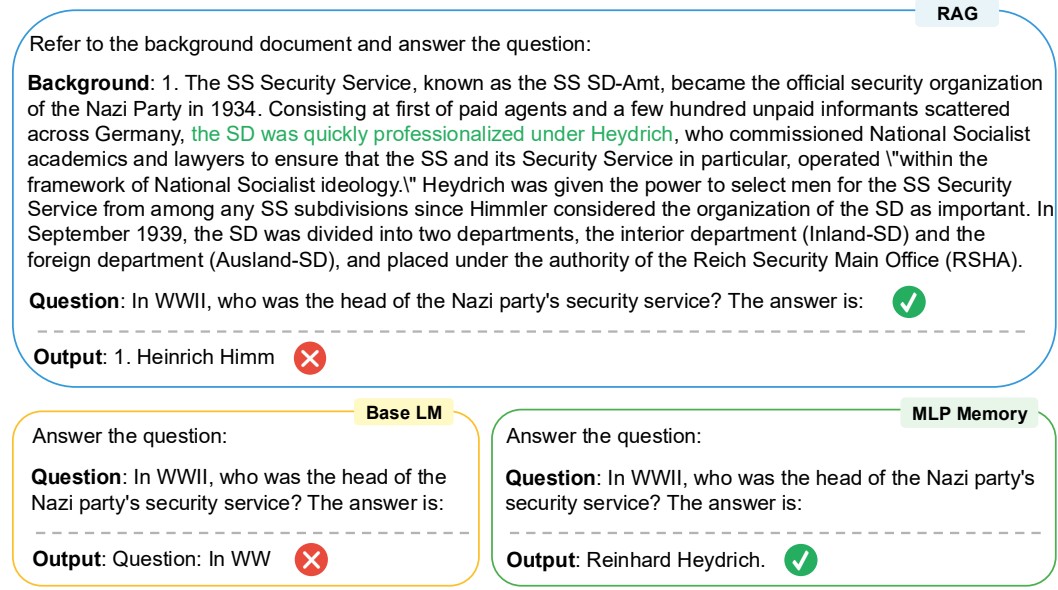

Figure 9: Comparison of model outputs on a factual question. Despite retrieving relevant documents with correct information (highlighted in green), RAG is misled by contextual distractors and produces an incorrect answer. MLP Memory generates the correct answer without explicit retrieval.

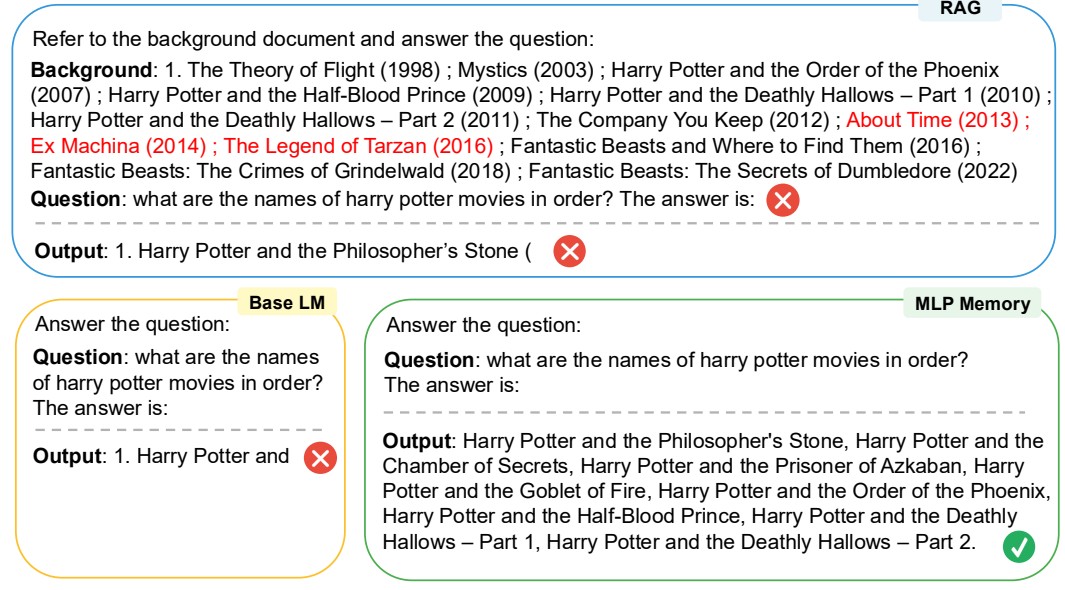

Figure 10: RAG retrieves irrelevant documents that introduce interference, while MLP Memory demonstrates perfect performance.

## J   CASE STUDY ON DOWNSTREAM TASKS

As show in Figure 9, we observe that RAG often fails even with relevant retrieved documents due to contextual noise interference. When documents contain related but distracting information, RAG's shallow integration cannot effectively filter these distractors, leading to incorrect answers. In contrast, MLP Memory learns intelligent corpus compression during training, capturing richer contextual relationships that enable robust disambiguation without explicit retrieval.

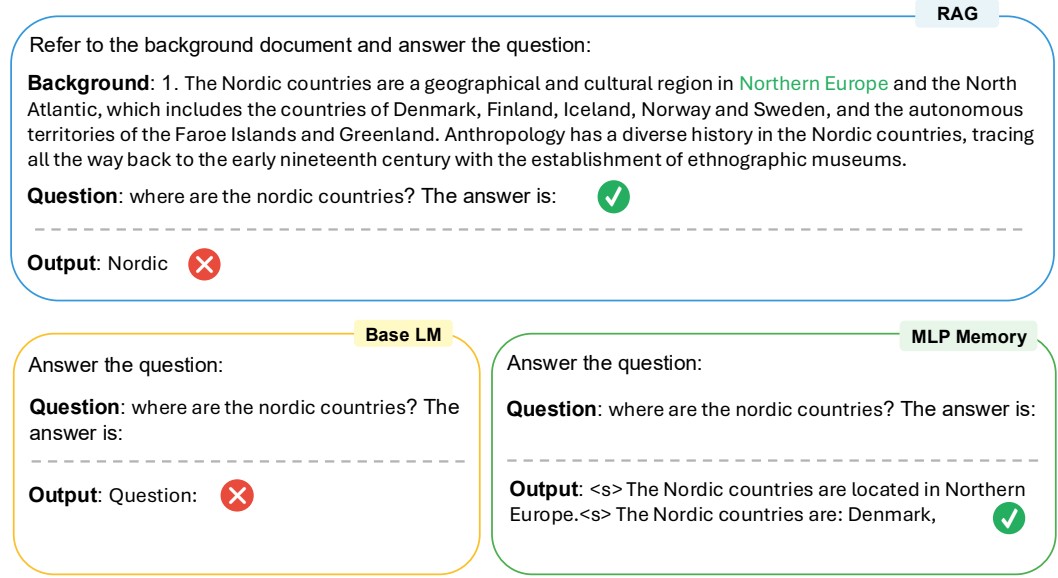

Figure 11: RAG fails to extract answer despite retrieving relevant content, while MLP Memory provides accurate response

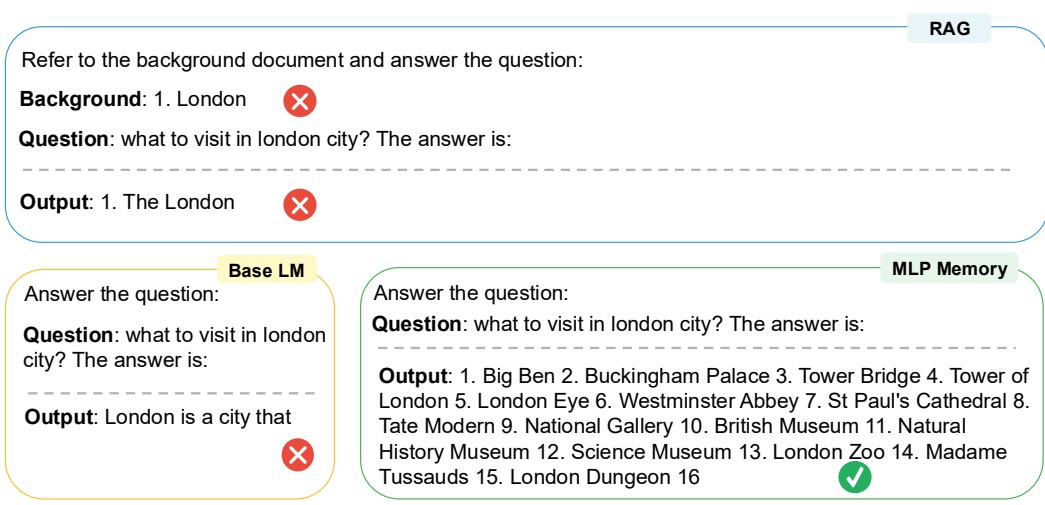

Figure 12: RAG system distracted by retrieved content fails to answer the question, while MLP Memory provides comprehensive and accurate response

# K    CASE STUDY ON THE DISTRIBUTION OF BASE LM, kNN-LM, AND MLP MEMORY

To further understand the mechanisms underlying the effectiveness of MLP Memory, we analyze the token-level probability distributions produced by the base LM, kNN-LM, and MLP Memory. Our hypothesis is that kNN-based distributions are particularly effective at capturing long-tail knowledge, whereas LM distributions exhibit strong coherence on frequent function words. MLP Memory appears to internalize both characteristics, acquiring long-tail information in a manner similar to non-parametric retrieval while preserving the stability of parametric models.

To examine this hypothesis, we perform case studies on examples drawn from WikiText-103 and report the probability assigned by each method to the target token (highlighted in bold) given its preceding context.

Table 10:   Token-level probability assignments for different methods on long-tail entities ( *top block* ) and coherent function words ( *bottom block* ).

| Long-tail Knowledge | | | |
|---|---|---|---|
| **Context (target token in bold)** | **Base LM** | **kNN-LM** | **MLP Memory** |
| Southward, in the Yongsan area, Keiser placed Brigadier General Joseph S. **Bradley**, Assistant Division Commander, in charge of the 9th Infantry Regiment. | 0.01 | 0.74 | **0.75** |
| The song reached number ten in **Mexico** and number one on both the Billboard Latin Songs and Latin Pop Songs chart. | 0.01 | 0.07 | **0.45** |
| Coherence | | | |
| **Context (target token in bold)** | **Base LM** | **kNN-LM** | **MLP Memory** |
| As the threat of invasion was clearly felt in late 1941, an idea for a series of secret observation posts (first in Gibraltar and later **in** other places like Malta and Aden)... | **0.65** | 0.01 | 0.44 |
| Here the invasion force encountered the first French defences, consisting of camouflaged trenches **and** pillboxes dug in along a ridge. | 0.45 | 0.06 | **0.53** |

As shown in Table 10, the case studies provide clear empirical support for our hypothesis. For rare entities such as *Bradley* and *Mexico*, MLP Memory assigns probabilities comparable to or exceeding those of kNN-LM, demonstrating effective acquisition of long-tail knowledge. In contrast, for function words such as *in* and *and*, MLP Memory maintains probability mass close to that of the base LM, whereas kNN-LM shows substantial degradation. These observations suggest that MLP Memory successfully combines the advantages of non-parametric retrieval with the coherence properties of parametric language models.

# L    MLP ARCHITECTURE DETAILS

The MLP Memory used in LLaMA2- and Mistral-based experiments is initialized from the corresponding MLP modules of their original architectures to maintain structural consistency and stable training behavior. Both settings employ 8 stacked MLP layers with their respective native hidden and intermediate dimensions. The total size of the MLP Memory is about 1B parameters, excluding embedding parameters. The detailed architectural configurations are reported in Table 11.

Table 11: MLP Memory architecture for LLaMA2 and Mistral experiments.

| Layers | Hidden dim | Intermediate dim |
|---|---|---|
| 8 | 4096 | 11008 |
| 8 | 4096 | 14336 |

In the study of the effect of MLP Memory size presented in Appendix M, the number of layers is varied while the hidden and intermediate dimensions are fixed at 1280 and 5120, respectively. The corresponding configurations are summarized in Table 12.

Table 12: MLP Memory size ablation configurations.

| Layers | Hidden dim | Intermediate dim |
|:------:|:----------:|:----------------:|
| 8 | 1280 | 5120 |
| 15 | 1280 | 5120 |
| 36 | 1280 | 5120 |

Based on the MLP Memory size study, we adopt the 8-layer configuration as the default setting, as it provides a favorable balance between performance and efficiency.

## M  EFFECT OF DIFFERENT MLP MEMORY SIZES ON PERFORMANCE

This section examines how different MLP Memory sizes affect language modeling performance on WikiText-103. The memory capacity is controlled by varying the number of MLP layers while keeping other architectural settings unchanged.

Table 13: Performance of different MLP Memory sizes on WikiText-103. The base model is GPT2-large.

| Model | WikiText-103 PPL |
|:------|:----------------:|
| GPT2-large | 15.81 |
| + MLP Memory (8 layers, 221M) | 11.41 |
| + MLP Memory (15 layers, 359M) | 11.35 |
| + MLP Memory (36 layers, 772M) | 11.25 |

As shown in Table 13, even the smallest MLP Memory achieves significant improvements over the base model. While scaling provides additional gains, smaller models offer the best performance-efficiency trade-off.

