# OpenReview forum: "MLP Memory: A Retriever-Pretrained Memory for Large Language Models"
_ICLR.cc/2026/Conference — ICLR 2026 Poster_

### Official Review · Reviewer_t9hD · 2025-10-27

**Soundness:** 3
**Presentation:** 3
**Contribution:** 2
**Rating:** 6
**Confidence:** 4

**Summary:**

The paper introduces MemoryMLP, a parametric module to mimic the behavior of explicit, external knowledge. In comparison to classical RAG approaches, MemoryMLP offers faster inference due to simple probability interpolation using the trained MLP module. The authors conduct extensive experiments showing that (1) MemoryMLP improves performance over base models and selected baselines (RAG / LoRA), (2) improves efficiency and throughput over these approaches, (3) reduces hallucination rates as measured by HaluEval, (4) ablations on the loss-function trade-off.

**Strengths:**

- The proposed idea of MemoryMLP is novel, simple and intuitive. It addresses a key limitation of many current RAG framework, overcoming search time and inference overhead. The presentation of the method including the respective image (Figure 4) is precise and easy to follow.
- The experiements show strong improvement over RAG and LoRA approaches, especially for Natural Questions and HotpotQA where the respective baseline do not improve on the base model performance. The choice of experiments is sound and convincing.
- The ablations and experiments on downstream tasks other than QA show the transferability of the proposed approach.

**Weaknesses:**

- The presentation may be contain flaws: While the overall presentation is good, I have stuggles on understanding several details of the method, either because the information is not where I would expect it to be or because it is simply not mentioned. For instance, some hyperparameters appear to be arbitrarily chosen such as the parameter count of the MLP or why the Wikipedia-2021 dump. Further, the MLP is not further specified, e.g., how many layers. See also the questions below.
- The results may be considered weak: While we observe relative improvements over the selected baselines, e.g., for NQ or HotpotQA are low and there are known to be existing approaches that are better. While state-of-the-art methods are not necessarily comparable, I would like to understand how MemoryMLP performs against other approaches such as question decomposition or fine-tuning a smaller LM. Especially considering the training budget of 5B token. See also question below.
- The evaluation may be considered incomplete: Do you also fine-tune the RAG baseline or do you employ them in a zero-shot setup? As proposed by Lewis et al. (2022), their RAG pipeline is also end-to-end differentiable and it would be interesting to see a comparison between MemoryMLP and end-to-end RAG.

**Questions:**

- The dataset construction in the main part is quite short - can you please specify how you create the embedding-distribution pairs? Specifically, how do ensure (i) that you generate useful examples such as in Figure 4 that actually depict a fact and (ii) that you have pair following identical format, also shown in Figure 4?
- Do you have any inights on performance differences using different MLP sizes? Why have you initially chosen 1B? How many layers have you chosen for the MLP?
- Do you think the training budget is better spend on optimizing a smaller LM on the target distribution? Why, why not?

---

> ### Author Response · Authors · 2025-11-21
> **Response to Reviewer t9hD (Part 1/2)**
>
> Thank you for the thoughtful feedback. We greatly appreciate the reviewer's recognition of our novel approach and strong experimental results.
>
> > Weakness 1: Presentation Issues
>
> We apologize for the lack of clarity regarding implementation details. We will provide comprehensive specifications in our revised manuscript. The MLP Memory uses an 8-layer architecture, with each layer following the design of the corresponding model (Llama or Mistral). The 1B parameter count is a direct consequence of this 8-layer configuration, which we empirically determined provides the optimal balance between performance and efficiency during preliminary research (see Part 2 for experimental results).
>
> Regarding the Wikipedia-2021 corpus selection, we followed the setting from Atlas [1], and Wikipedia-2021 is the newest corpus available in its repo.
>
> > Weakness 2: Additional Baselines
>
> We appreciate your suggestion for additional comparisons. Following your recommendation, we evaluated our method against the recent query decomposition approach Least-to-Most Prompting [2]:
>
> | Method | NQ | WebQA | TriviaQA | TruthfulQA | HotpotQA | AVG |
> |--------|------|--------|----------|------------|----------|------|
> | Llama2 | 23.18 | 32.09 | 56.91 | 29.16 | 22.72 | 32.81 |
> | +Least-to-Most | **27.78** | 34.65 | 49.52 | 29.68 | 24.19 | 33.16 |
> | +MLP Mem | 27.04 | **36.61** | **57.50** | **30.04** | **25.69** | **35.38** |
> | Mistral | 20.63 | 29.28 | 57.65 | 32.09 | 20.96 | 32.12 |
> | +Least-to-Most | 25.19 | **42.03** | 53.10 | 32.15 | 20.83 | 34.66 |
> | +MLP Mem | **25.20** | 37.45 | **60.99** | **32.54** | **24.14** | **36.06** |
>
> Our results demonstrate that MLP Memory outperforms the Least-to-Most prompting method across most datasets.
>
> We want to emphasize that all benchmarks are evaluated in a zero-shot setting with MLP Memory. Our objective is to compress the **entire pre-training corpus** into a parametric model. Thus, MLP Memory should be considered orthogonal to fine-tuning approaches, which fall outside the scope of our comparative analysis.
>
> > Weakness 3: End-to-end RAG Comparison
>
> We appreciate the reviewer's suggestion to compare against end-to-end differentiable RAG methods. To ensure fair comparison, we employ RAG in a zero-shot configuration throughout our experiments. While the original RAG framework [3] proposes joint training of the generator and retriever, the computational requirements of joint fine-tuning the 7B generator model exceed our available resources. Consequently, we evaluate against REPLUG [4], a more recent end-to-end training approach.
>
> Following the REPLUG LSR training protocol, we fine-tune the BGE retriever model on 300k sequences of 256 tokens sampled from the WikiText-103 dataset. Each sequence comprises 128 tokens of input context followed by 128 tokens of ground truth continuation. We adopt the hyperparameters specified in the original REPLUG paper and train for 2 epochs, yielding the following results:
>
> | Mistral      | NQ    | WebQA | TriviaQA | TruthfulQA | HotpotQA | AVG   |
> |--------------|-------|-------|----------|------------|----------|-------|
> | base         | 20.63 | 29.28 | 57.65    | 32.09      | 20.96    | 32.12 |
> | +Vanilla RAG | 22.56 | 24.90 | 54.21    | **35.47**      | **29.77**    | 33.38 |
> | +REPLUG      | 22.14 | 26.75 | 56.28    | 33.36      | 25.44    | 32.79 |
> | +MLP Mem     | **25.20** | **37.45** | **60.99**    | 32.54      | 24.14    | **36.06** |
>
> The results indicate that MLP Memory outperforms REPLUG across three datasets (NQ, WebQA, and TriviaQA), though REPLUG maintains a slight advantage on TruthfulQA and HotpotQA. Our findings align with the reproduction results reported in RetroLLM [5], demonstrating that REPLUG underperforms vanilla RAG on HotpotQA while achieving comparable performance on other datasets. We hypothesize that this performance gap stems from a task format mismatch: the REPLUG LSR training objective may not effectively transfer to the question answering evaluation paradigm.
>
> > References:
>
> [1] Izacard, Gautier, et al. "Atlas: Few-shot learning with retrieval augmented language models." Journal of Machine Learning Research 24.251 (2023): 1-43.
>
> [2] Zhou, Denny, et al. "Least-to-most prompting enables complex reasoning in large language models." arXiv preprint arXiv:2205.10625 (2022).
>
> [3] Lewis, Patrick, et al. "Retrieval-augmented generation for knowledge-intensive nlp tasks." Advances in neural information processing systems 33 (2020): 9459-9474.
>
> [4] Shi, Weijia, et al. "Replug: Retrieval-augmented black-box language models." Proceedings of the 2024 Conference of the North American Chapter of the Association for Computational Linguistics: Human Language Technologies (Volume 1: Long Papers). 2024.
>
> [5] Li, Xiaoxi, et al. "Retrollm: Empowering large language models to retrieve fine-grained evidence within generation." Proceedings of the 63rd Annual Meeting of the Association for Computational Linguistics (Volume 1: Long Papers). 2025.

---

> > ### Author Response · Authors · 2025-11-21
> > **Response to Reviewer t9hD (Part 2/2)**
> >
> > > Question 1: Dataset construction
> >
> > We apologize for the confusion. The examples in Figure 4 are purely illustrative to aid understanding of the pipeline. In practice, we construct embedding-distribution pairs for **every token in the pre-training corpus**. This approach ensures that MLP Memory's pre-training is fully self-supervised, requires no additional annotations, thus enables seamless scaling.
> >
> > > Question 2: Scaling of MLP Memory
> >
> > We conducted preliminary experiments with different MLP scales on WikiText-103:
> >
> > | Method | WikiText-103 PPL |
> > |--------|-----------------|
> > | gpt2-large | 15.81 |
> > | +MLP Memory-small (8 layers) | 11.41 |
> > | +MLP Memory-med (15 layers) | 11.35 |
> > | +MLP Memory-large (36 layers) | 11.25 |
> >
> > While additional layers provide marginal improvements, the 8-layer configuration achieves the best performance-efficiency tradeoff. This architecture choice results in approximately 1B parameters for Llama and Mistral models.
> >
> > > Question 3: Decoder-like Memory
> >
> > We compared MLP Memory with a Decoder model of equivalent parameter count trained using the same objective:
> >
> > | Method | PPL |
> > |--------|-----|
> > | gpt2-large | 15.81 |
> > | +MLP Memory-large | 11.25 |
> > | +Decoder Memory-large | 11.52 |
> >
> > While the Decoder Memory also shows significant improvement, MLP Memory performs better. We hypothesize that the training objective (single embedding → distribution) is particularly more suited for MLP architectures, as no token-mixing operation (Attention) is required.

---

### Official Review · Reviewer_qhzi · 2025-11-01

**Soundness:** 2
**Presentation:** 3
**Contribution:** 3
**Rating:** 4
**Confidence:** 4

**Summary:**

This paper introduces a new method called MLP Memory to help large language models be more factually accurate without slowing them down.  The proposed solution involves pre-training a lightweight, all-MLP (Multi-Layer Perceptron) module to imitate the output distribution of a kNN-LM retriever over the entire pre-training corpus. This MLP Memory module learns to map a hidden state from the LLM to a vocabulary distribution that approximates the retriever's output, effectively internalizing retrieval patterns. During inference, the MLP Memory's distribution is interpolated with the base LLM's output, providing a knowledge-augmented prediction. The results show that this method makes models better at answering questions, reduces hallucinations, and is much faster than using a traditional search (RAG).

**Strengths:**

(1) The motivation of the paper is clear. The paper identifies a clear gap in the literature and proposes an elegant hybrid approach that bridges parametric and non-parametric methods.

(2) Comprehensive evaluation: The experimental design is thorough, evaluating the method across multiple critical dimensions: factual question answering, general NLP capability preservation, hallucination reduction, and inference efficiency.

(3) Compelling performance: The results show that MLP Memory not only surpasses the base models but also outperforms RAG on accuracy metrics. This suggests the method successfully captures richer contextual relationships. Results also show that MLP Memory gets better efficiency than existing methods.

**Weaknesses:**

(1)  The method requires a pre-training phase for the MLP Memory module, which involves constructing a large datastore and optimizing a large model. The computational cost and time of this initial step could be a barrier for some.

(2)  Unlike RAG, the knowledge stored in the MLP Memory is fixed after pre-training (just like the generative retrieval models). This limits the model's ability to access updated or real-time information without a costly retraining cycle, making it less suitable for domains requiring frequent knowledge updates.

(3) While the paper demonstrates that the method works, a deeper analysis of what specific knowledge patterns the MLP Memory learns and how it interfaces with the transformer's inherent reasoning process would strengthen the theoretical contribution.

(4) Some existing works are trying to solve similar problems. For example, RetroLLM also tries to leverage generative retrieval models to implement end-to-end RAG systems. The authors should discuss and compare with these existing methods.

[1] RetroLLM: Empowering Large Language Models to Retrieve Fine-grained Evidence within Generation, Li et al..

**Questions:**

(1) Cost problem: What is the total computational budget (e.g., in GPU hours) required to pre-train the 1B-parameter MLP Memory for a 7B model? How does this cost scale with the size of the base model and the pre-training corpus?

(2) Dynamic knowledge integration: Could the MLP Memory architecture be adapted for incremental learning? For instance, is it feasible to fine-tune the MLP Memory on a small corpus of new documents to update its knowledge without catastrophic interference?

(3) Generalization to specialized domains: The experiments use general corpora like Wikipedia. How would the method perform when applied to highly specialized domains (e.g., legal or biomedical text) where the retrieval might be more complex? For example, in the legal domain, retrieving similar legal cases for generative legal judgment prediction is much difficult than the general RAG systems.

---

> ### Author Response · Authors · 2025-11-21
> **Response to Reviewer qhzi (Part 1/2)**
>
> Thank you for the thoughtful feedback. We greatly appreciate the reviewer's recognition of our clear motivation, comprehensive experimental evaluation, and compelling empirical results demonstrating both performance gains and efficiency improvements.
>
> > Weakness 1 & Question 1: Training Cost
>
> We provide a detailed breakdown of the computational requirements to clarify the training cost of our approach. The training pipeline consists of three components: (1) 7B model inference on the training dataset to obtain embeddings, (2) kNN search for each training token's embedding, and (3) MLP Memory pre-training.
>
> Our analysis on an 8-node, 8×A800 configuration yields the following measurements:
>
> | Component | GPU hours |
> |-----------|-----------|
> | 7B model 1 epoch CPT | 12h |
> | 7B model inference | 4h |
> | KNN search | 8h |
> | MLP Memory training | 12h |
>
> The total training cost for MLP Memory is equivalent to 2 epochs of continued pre-training for the 7B model. Given the substantial performance and efficiency improvements, we believe this computational investment is well justified.
>
> Regarding scalability, we clarify two aspects. First, the training cost scales linearly with the pretraining corpus size just like traditional LM pre-training. Second, among the three training components, only the one-time inference pass over the training dataset depends on the base model. This step constitutes only 16% of the total training cost, and the other two steps, kNN search and MLP Memory training are independent of the base model size.
>
> > Weakness 2 & Question 2: Knowledge Updating
>
> We conducted experiments to demonstrate MLP Memory's capability for incremental learning. We sampled two non-overlapping shards (shard A and shard B) from the Wikipedia corpus, each containing approximately 5% of the original data, and performed sequential training:
>
> | | PPL | NQ | HotpotQA | WebQA | TriviaQA | TruthfulQA | AVG |
> |---|---|---|---|---|---|---|---|
> | Mistral | 9.84 | 20.63 | 20.96 | 29.28 | 57.65 | 32.09 | 32.12 |
> | +shard A | 8.51 | 24.65 | 24.18 | 35.09 | 58.20 | 32.32 | 34.89 |
> | +shard A & shard B | 8.29 | 24.08 | 23.94 | 36.27 | 58.79 | 32.32 | 35.08 |
>
> The results show consistent improvements in both perplexity and QA performance across incremental training stages, with no evidence of catastrophic forgetting. This demonstrates that MLP Memory can efficiently incorporate new knowledge through continued training on additional data.
>
> We acknowledge that knowledge editing remains challenging for parametric models, including both MLP Memory and generative retrieval method. Nevertheless, our approach delivers substantial practical advantages: over 12% improvement on QA tasks, 10+ points gain on HaluEval, 7× faster inference compared to kNN-LM, and a remarkable 10,000× compression ratio (40TB datastore compressed to 4GB).
>
> > Weakness 3: Deeper Analysis
>
> We conducted analysis to elucidate the mechanisms underlying MLP Memory's effectiveness. Our hypothesis is that kNN distributions excel at capturing long-tail knowledge while LM distributions maintain coherence, and that MLP Memory successfully internalizes both characteristics—acquiring long-tail knowledge similar to non-parametric methods while preserving the coherence of parametric models.
>
> To validate this hypothesis, we performed case studies on the wikitext-103 dataset. We report the probability assigned by each method to the target token (highlighted) given the preceding context:
>
> **Long-tail Knowledge:**
>
> | Context | Base | kNN-LM | MLP Memory |
> |---------|------|--------|------------|
> | Southward, in the Yongsan area, Keiser placed Brigadier General Joseph S. ***Bradley***, Assistant Division Commander, in charge of the 9th Infantry Regiment.. | 0.01 | 0.74 | **0.75** |
> | The song reached number ten in ***Mexico*** and number one on both the Billboard Latin Songs and Latin Pop Songs chart. | 0.01 | 0.07 | **0.45** |
>
> **Coherence:**
>
> | Context | Base | kNN-LM | MLP Memory |
> |---------|------|--------|------------|
> | As the threat of invasion was clearly felt in late 1941, an idea for a series of secret observation posts (first in Gibraltar and later ***in*** other places like Malta and Aden)... | **0.65** | 0.01 | 0.44 |
> | Here the invasion force encountered the first French defences, consisting of camouflaged trenches ***and*** pillboxes dug in along a ridge. | 0.45 | 0.06 | **0.53** |
>
> The case studies provide strong empirical support for our hypothesis. For rare entities like "Bradley" and "Mexico," MLP Memory achieves probability assignments comparable to or exceeding kNN-LM, demonstrating effective acquisition of long-tail knowledge. For function words like "in" and "and," MLP Memory maintains reasonable probability mass relative to the base model, whereas kNN-LM shows severe degradation (0.01 and 0.06). These results confirm that MLP Memory successfully combines the knowledge retrieval capabilities of non-parametric methods with the coherence of parametric models.

---

> > ### Author Response · Authors · 2025-11-21
> > **Response to Reviewer qhzi (Part 2/2)**
> >
> > > Weakness 4: End-to-end RAG Systems
> >
> > We thank the reviewer for highlighting the innovative work of RetroLLM [1], which we will add to our related works. Both RetroLLM and MLP Memory aim to unify retrieval and generation, though they adopt different approaches.
> >
> > **Training Pipeline:** MLP Memory features a simpler and more end-to-end training framework. Our training objective directly learns to predict the kNN distribution from current token embeddings. In contrast, inspired by DSI [2], RetroLLM employs constrained decoding to generate clues and evidence. This requires more complex training procedures and data preparation, including the construction of hierarchical FM-Index constraints and implementation of forward-looking decoding strategies.
> >
> > **Inference Efficiency:** RetroLLM provides greater transparency by generating explicit evidence along the generation path, which aids interpretability. However, this comes at the cost of increased Time To First Token (TTFT), as the model must generate intermediate clues and evidence before producing the final answer. MLP Memory avoids this overhead by requiring only a single forward pass through a lightweight MLP architecture.
> >
> > Both methods represent valuable contributions to unified retrieval-generation frameworks, with RetroLLM emphasizing evidence transparency while MLP Memory prioritizes inference efficiency and deployment simplicity.
> >
> > > Question 3: Generalization to Specialized Domains
> >
> > We evaluated MLP Memory on the legal domain using the CaseHOLD and SCOTUS dataset following the setting from [3], with holdings from the training split as our pre-training corpus:
> >
> > | | case_hold (micro_F1) | case_hold (macro_F1) | scotus(micro_F1) | scotus(macro_F1) |
> > |---|---|---|---|---|
> > | mistral | 35.72 | 35.73 | 26.67 | 13.32 |
> > | +kNN-LM | 35.81 | 35.82 | 25.00 | 11.60 |
> > | +MLP Memory(0.5B) | **36.52** | **36.53** | **30.0** | **24.15** |
> >
> > MLP Memory demonstrates consistent improvements across all metrics. These results showcase the method's ability to generalize to specialized domains with distinct linguistic patterns and domain-specific knowledge, confirming its applicability beyond general knowledge tasks.
> >
> > > References:
> >
> > [1] Li, Xiaoxi, et al. "Retrollm: Empowering large language models to retrieve fine-grained evidence within generation." Proceedings of the 63rd Annual Meeting of the Association for Computational Linguistics (Volume 1: Long Papers). 2025.
> >
> > [2] Tay, Yi, et al. "Transformer memory as a differentiable search index." Advances in Neural Information Processing Systems 35 (2022): 21831-21843.
> >
> > [3] Cheng, Daixuan, Shaohan Huang, and Furu Wei. "Adapting large language models via reading comprehension." The Twelfth International Conference on Learning Representations. 2023.

---

### Official Review · Reviewer_hdzS · 2025-11-01

**Soundness:** 3
**Presentation:** 2
**Contribution:** 3
**Rating:** 6
**Confidence:** 4

**Summary:**

This paper introduces MLP Memory, a parametric memory module trained to imitate the behavior of retrieval-augmented models such as kNN-LM. The approach constructs a datastore from a training corpus and trains a small MLP to predict the kNN distribution given hidden representations from a pretrained language model. During inference, the predicted distribution is interpolated with the base model’s output probabilities, removing the need for explicit retrieval or datastore lookup. Experiments on question answering and factual recall tasks show that MLP Memory achieves comparable or better accuracy than kNN-LM and RAG while being more efficient in inference, as measured by time-to-first-token (TTFT) and tokens-per-second (TPS). Ablations explore the layer position for attaching the memory and the balance between cross-entropy and KL objectives.

**Strengths:**

The paper addresses an important problem in retrieval-augmented modeling: maintaining factual recall benefits without external retrieval or large storage. MLP Memory provides a clean parametric alternative that approximates kNN-LM behavior through distribution imitation. The method is conceptually clear and empirically well evaluated, with strong results on factual QA and hallucination benchmarks. The use of TTFT and TPS as efficiency metrics is appropriate and provides a runtime-based comparison to retrieval baselines. Ablation studies on interpolation weighting, attachment layer depth, and loss balancing help clarify key design choices. Overall, the work presents a technically sound and well-executed exploration of parametric memory.

**Weaknesses:**

Despite strong results, several aspects limit the clarity and generality of the approach. The training cost is not clearly quantified: even if inference is faster, pretraining the MLP Memory requires generating kNN distributions and performing supervised training on them, which likely increases per-step computation and total cost. It is also unclear whether the backbone model is frozen or jointly updated, which affects the practical difficulty of integration. Moreover, the MLP Memory seems to require task-specific pretraining, since the datastore and supervision are derived from each corpus; this reduces its plug-and-play appeal compared to non-parametric retrieval systems like RAG. The method also depends on tuning the interpolation weight λ for each task, which introduces additional hyperparameter sensitivity. Finally, while the ablation identifies an optimal attachment depth (~70%), it remains uncertain how this generalizes across different architectures or scales. These issues make the approach less universally efficient than implied.

**Questions:**

- Is MLP Memory pretrained separately for each task or domain, or can a single memory generalize across corpora?
- Are the base language model weights frozen during memory pretraining, or do they also receive updates?
- How large is the total training cost (in GPU-hours or FLOPs) relative to standard fine-tuning or RAG adaptation?
- How sensitive are results to the choice of $\lambda$, and how is it tuned for each task?
- What is the scaling trend with different memory sizes? Does a smaller MLP maintain performance efficiency?
- Could you discuss missing important related literature on (efficient) parametric memory architectures such as Product Key Memory [1, 2] and Entities as Experts [3, 4].

[1] Lample et al., Large Memory Layers with Product Keys

[2] Kim and Jung, Large Product Key Memory for Pretrained Language Models

[3] Fevry et al., Entities as Experts: Sparse Memory Access with Entity Supervision

[4] Verga et al., Facts as Experts: Adaptable and Interpretable Neural Memory over Symbolic Knowledge

---

> ### Author Response · Authors · 2025-11-21
> **Response to Reviewer hdzS (Part 1/2)**
>
> Thank you for the thoughtful feedback. We greatly appreciate the reviewer's recognition of our novel approach and strong empirical results.
>
> > Weakness 1 & Question 3: Quantification of training cost
>
> We have meticulously analyzed the training cost, dividing it into three components: (1) 7B model inference on the training dataset for embeddings, (2) kNN search for each training token's embedding, and (3) MLP Memory pre-training.
> Our analysis on an 8-node, 8×A800 setup shows:
>
> | Component | GPU hours |
> |-----------|-----------|
> | 7B model 1 epoch CPT | 12h |
> | 7B model inference | 4h |
> | KNN search | 8h |
> | MLP Memory training | 12h |
>
> The total MLP Memory training cost equals merely 2 epochs of CPT for the 7B model, which we believe is reasonable given the substantial performance gains achieved. We also emphasize that our pre-training is self-supervised rather than supervised, as the supervision signals are self-generated without requiring additional labeling, making our framework particularly suitable for future scaling.
>
> > Weakness 2 & Question 2: Base model frozen or jointly trained
>
> The base model remains frozen during MLP Memory pre-training, which significantly reduces training costs. While joint training presents exciting potential for future improvements, keeping the model frozen during our current approach ensures practical deployability.
>
> > Weakness 3: Plug-and-play capabilities
>
> We want to clarify that MLP Memory pre-training operates on general corpora rather than task-specific datasets. Pre-trained on the entire Wikipedia corpus, it generalizes to all tasks requiring knowledge from that corpus. For a specific model, our method's plug-and-play capability matches that of traditional RAG systems. However, we acknowledge that we need to train different MLP Memory for different models.
>
> > Weakness 4 & Question 4: Sensitivity of interpolation weight λ
>
> The sensitivity analysis for interpolation weight λ is provided in Appendix D. Our experiments demonstrate that performance remains stable within the 0.3-0.6 range, with only gradual degradation outside this interval. This confirms that our method is robust to hyperparameter choice within reasonable ranges, facilitating practical deployment without extensive tuning. For each task, we tune λ on the validation dataset following the established methodology in [1].
>
> > Weakness 5: Optimal attachment depth
>
> As detailed in Section 5.4, paragraph 3, MLP Memory consistently achieves optimal performance at 70-80% of network depth across GPT-2 small, medium, and large scales. This pattern holds for Llama and Mistral models as well. Importantly, using layers beyond 80% show minimal degradation (less than 8% perplexity increase), demonstrating the robustness of our approach.
>
> > References
>
> [1] Shi, Weijia, et al. "knn-prompt: Nearest neighbor zero-shot inference." arXiv:2205.13792 (2022)

---

> > ### Author Response · Authors · 2025-11-21
> > **Response to Reviewer hdzS (Part 2/2)**
> >
> > > Question 1: Generalization ability
> >
> > One MLP Memory successfully generalizes to all tasks within its training corpus. Our experiments show that a single MLP Memory pre-trained on the entire Wikipedia corpus (5B tokens) generalizes effectively across all Wikipedia-based tasks. While computational constraints currently limit us to this scale, we believe the generalization potential extends further. Scaling MLP Memory to match modern LLMs' pre-training corpora (>300B tokens) remains an exciting direction for future work.
> >
> > > Question 5: Scaling Trend
> >
> > Following your valuable suggestion, we evaluated different MLP Memory scales:
> >
> > | Model | WikiText-103 PPL |
> > |-------|-----------------|
> > | GPT2-large | 15.81 |
> > | +MLP Memory-small (8 layers) | 11.41 |
> > | +MLP Memory-med (15 layers) | 11.35 |
> > | +MLP Memory-large (36 layers) | 11.25 |
> >
> > Results show that even the smallest MLP Memory achieves significant improvements over the base model. While scaling provides additional gains, smaller models offer the best performance-efficiency trade-off.
> >
> > > Question 6: Related literature
> >
> > We appreciate the reviewer's suggestion to discuss related work more comprehensively. Our method represents a novel dense memory approach, distinct from the sparse memory architectures in the suggested literature:
> >
> > Compared to Product Key Memory methods [1,2], Entities as Experts [3,4] and recent works like UltraMemory [5,6], our dense approach offers several advantages:
> >
> > 1. **Memory efficiency**: While sparse methods activate fewer parameters, they still require loading all memory slots into RAM. For example, [5] uses 20 million memory slots, creating substantial memory overhead despite sparse activation.
> >
> > 2. **Training stability**: Sparse memory architectures face the training instability and load balancing challenges inherent to extreme MoE architectures, as acknowledged in [5]. Our dense approach avoids these issues entirely.
> >
> > 3. **Self-supervised training**: Unlike Entities as Experts [3,4], which require entity linking during data preparation, MLP Memory's pre-training is fully self-supervised, requiring no additional annotations and enabling seamless scaling.
> >
> > We acknowledge that sparse memory offers advantages in knowledge updating and inference efficiency through minimal parameter activation. However, our work establishes a new paradigm for dense memory architectures that warrants further exploration alongside sparse approaches.
> >
> > > References
> >
> > [1] Lample, Guillaume, et al. "Large memory layers with product keys." Advances in Neural Information Processing Systems 32 (2019).
> >
> > [2] Kim, Gyuwan, and Tae Hwan Jung. "Large product key memory for pretrained language models." Findings of the Association for Computational Linguistics: EMNLP 2020. 2020.
> >
> > [3] Févry, Thibault, et al. "Entities as experts: Sparse memory access with entity supervision." arXiv preprint arXiv:2004.07202 (2020).
> >
> > [4] Verga, Pat, et al. "Facts as experts: Adaptable and interpretable neural memory over symbolic knowledge." arXiv preprint arXiv:2007.00849 (2020).
> >
> > [5] Huang, Zihao, et al. "Ultra-sparse memory network." arXiv preprint arXiv:2411.12364 (2024).
> >
> > [6] Huang, Zihao, et al. "UltraMemV2: Memory Networks Scaling to 120B Parameters with Superior Long-Context Learning." arXiv preprint arXiv:2508.18756 (2025).

---

### Official Review · Reviewer_xCCs · 2025-11-01

**Soundness:** 2
**Presentation:** 3
**Contribution:** 2
**Rating:** 4
**Confidence:** 2

**Summary:**

This paper proposes MLP-memory that replaces the KV store in kNN-LM by an MLP. The goal is to save memory space requirement, which is high in kNN-LM. The problem is interesting. A more efficient implementation of kNN-LM idea can make the approach more feasible in practice.
The method is tested on several tasks of QA and general NLP, showing better performance compared to the baselines. Analyses are provided on the impact of different hyperparameters and the training objective function.

**Strengths:**

1. The main idea is to make kNN-LM more storage efficient. It trains an MLP to replace the costly K-V storage. The idea is interesting.

2. The paper describes well the approach. It contains good analyses about different hyperparameters and settings.

3. The approach is compared to the relevant baselines (except the missing comparison mentioned in weakness).

**Weaknesses:**

1. The contribution of the paper is limited. It does not change the basic idea of kNN-LM, but tries to provide a better implementation of it.

2. As the main contribution of MLP-memory is to reduce the storage cost of kNN-LM, the main comparison should be done with kNN-LM. However, in the main results on QA, this comparison is missing. The comparison is only done on general NLP tasks. This is insufficient.

3. The goal of MLP-memory is to reproduce kNN-LM at lost storage cost. The key research question is whether MLP can approximate KV store. This question requires extensive comparison between them. In addition to the end tasks, it is also useful to compare the internal probability distribution of models (e.g. PPL). The paper contains some analysis about the distribution of probabilities of LM, kNN-LM and MLP-memory, but perplexity is only evaluated on MLP-memory. A comparison on perplexity may better reveal whether MLP-memory can approach kNN-LM.

4. The analysis (in appendix) on the different distributions of probabilities in LM, kNN-LM and MLP-memory is interesting, showing that MLP-memory is between LM and kNN-LM. It is however unclear what the implication would be. So the authors suggest that a distribution between LM and kNN-LM sould be better? and why?

**Questions:**

see weakness.

---

> ### Author Response · Authors · 2025-11-21
> **Response to Reviewer xCCs (Part 1/2)**
>
> Thank you for the thoughtful feedback. We greatly appreciate the reviewer's recognition of our novel approach and comprehensive analysis.
>
> > Weakness 1: Limited Contribution
>
> We respectfully disagree that MLP Memory is merely a better implementation of kNN-LM. Our approach represents a fundamental paradigm shift from traditional retrieval to learning a parametric memory through retrieval. The non-parametric, dense retrieval is fundamentally limited by its vector-matching principle, impeding its ability to capture complex forms of knowledge, and its non-parametric nature disables end-to-end training of RAG systems. On the contrary, MLP Memory learns to memorize by compressing knowledge into parametric neural networks. The parametric MLP memory is pretrained by retrieval imitation and next token prediction, not only enabling end-to-end training, but also exceeds the limitation of explicit vector-matching. This also explains why our MLP Memory consistently outperforms kNN-LM in our experiments.
>
> To draw an analogy, neural language models are not merely better implementations of n-gram models; they represent a fundamentally different approach that dramatically outperforms their predecessors, despite using the same next-token-predictoin task. Similarly, MLP Memory removes explicit retrieval, circumvents all of its downsides (slow retrieval, non-parametric nature disabling end-to-end optimization) while still being able to exceed retrieval-based methods in terms of performance.
>
> Our experimental results strongly support this distinction:
>
> 1. **Substantial Performance Gains**: MLP Memory achieves 12%+ improvement on QA tasks and 10%+ on HaluEval, significantly outperforming kNN-LM across all benchmarks.
>
> 2. **Superior Efficiency**: Beyond performance gains, MLP Memory delivers 7× faster inference (TPS) compared to kNN-LM while achieving 10,000× compression ratio (40TB datastore → 4GB model).
>
> These results demonstrate that MLP Memory is not simply an incremental improvement but a fundamentally new approach to knowledge representation in language models.
>
> > Weakness 2: kNN-LM QA Performance
>
> Thank you for raising this important point. We have conducted additional experiments comparing kNN-LM performance:
>
> | Method | NQ | WebQA | TriviaQA | TruthfulQA | HotpotQA | AVG |
> |--------|-------|--------|-----------|-------------|----------|-------|
> | Llama2 | 23.18 | 32.09 | 56.91 | 29.16 | 22.72 | 32.81 |
> | +kNN-LM | 23.16 | 33.46 | 57.31 | 29.22 | 22.66 | 33.16 |
> | +MLP Mem | **27.04** | **36.61** | **57.50** | **30.04** | **25.69** | **35.38** |
> | Mistral | 20.63 | 29.28 | 57.65 | 32.09 | 20.96 | 32.12 |
> | +kNN-LM | 21.05 | 30.51 | 57.77 | 32.33 | 21.20 | 32.57 |
> | +MLP Mem | **25.20** | **37.45** | **60.99** | **32.54** | **24.14** | **36.06** |
>
> We initially excluded kNN-LM comparisons based on findings from [1], which reported performance degradation on knowledge-intensive tasks like NQ and HotpotQA across three different models when equipped with kNN-LM. In our experiments, the larger Wikipedia datastore enables kNN-LM to achieve marginal improvements over the base model—a notable contrast to the performance drops reported in [1], though the gains remain modest.
>
> MLP Memory, in contrast, delivers substantial improvements across all benchmarks without the reasoning degradation typically associated with retrieval methods. This demonstrates a critical advantage: our approach successfully enhances memorization through learned retrieval patterns while preserving the model's reasoning capabilities—addressing a fundamental limitation that has plagued traditional retrieval-augmented approaches.
>
> > References:
>
> [1] Geng, Shangyi, Wenting Zhao, and Alexander M. Rush. "Great Memory, Shallow Reasoning: Limits of $k$NN-LMs." arXiv preprint arXiv:2408.11815 (2024).

---

> > ### Author Response · Authors · 2025-11-21
> > **Response to Reviewer xCCs (Part 2/2)**
> >
> > > Weakness 3: Perplexity Evaluation
> >
> > Thank you for this suggestion. We provide perplexity evaluations below:
> >
> > | **Llama 2** | ppl |
> > |-------------|------|
> > | base | 10.05 |
> > | +kNN-LM | 6.99 |
> > | +MLP Memory | **6.97** |
> >
> > | **Mistral** | ppl |
> > |-------------|------|
> > | base | 9.84 |
> > | +kNN-LM | 7.72 |
> > | +MLP Memory | **7.27** |
> >
> > MLP Memory surpasses kNN-LM in perplexity, particularly for Mistral.
> >
> > > Weakness 4: Distribution Analysis
> >
> > Thank you for your thorough examination of our appendix. Our preliminary analysis reveals that kNN distributions demonstrate superior performance on long-tail knowledge, while LM distributions excel at maintaining coherence. We hypothesize that MLP Memory can effectively internalize long-tail knowledge (analogous to non-parametric methods) while preserving the coherence characteristic of parametric models. Specifically, we expect MLP Memory to exhibit kNN-like distributions for long-tail knowledge tokens and LM-like distributions for function tokens and continuation tokens.
> >
> > To empirically validate this hypothesis, we conducted case studies on the WikiText-103 corpus, examining two critical dimensions (we report the label probability assigned by each method to the highlighted token, conditioned on the preceding context):
> >
> > **Long-tail knowledge:**
> >
> > | Context | Base | kNN-LM | MLP Memory |
> > |---------|------|--------|------------|
> > | Southward, in the Yongsan area, Keiser placed Brigadier General Joseph S. ***Bradley***, Assistant Division Commander, in charge of the 9th Infantry Regiment.. | 0.01 | 0.74 | **0.75** |
> > | The song reached number ten in ***Mexico*** and number one on both the Billboard Latin Songs and Latin Pop Songs chart. | 0.01 | 0.07 | **0.45** |
> >
> > **Coherence:**
> >
> > | Context | Base | kNN-LM | MLP Memory |
> > |---------|------|--------|------------|
> > | As the threat of invasion was clearly felt in late 1941, an idea for a series of secret observation posts (first in Gibraltar and later ***in*** other places like Malta and Aden)... | **0.65** | 0.01 | 0.44 |
> > | Here the invasion force encountered the first French defences, consisting of camouflaged trenches ***and*** pillboxes dug in along a ridge. | 0.45 | 0.06 | **0.53** |
> >
> > Our case studies provide strong empirical support for the proposed hypothesis. For rare entities such as "Bradley" and "Mexico," MLP Memory achieves probability assignments comparable to or exceeding those of kNN-LM, demonstrating effective knowledge acquisition. For function words such as "in" and "and," MLP Memory maintains reasonable probability mass relative to the base model, whereas kNN-LM exhibits severe degradation (0.01 and 0.06).

---

### Author Response · Authors · 2025-11-21
**Summary of Revisions**

We thank all reviewers for their helpful and constructive feedback. In the revised PDF, we have made the following changes:
- Section 4: In paragraph 2 of the Implementation Details, we now explicitly state the number of layers used in the MLP.
- Appendix I: We clarify that the model previously referred to as “GPT2-large” actually denotes “GPT2-large-CPT,” i.e., GPT2-large with continued pre-training on WikiText-103, and we adjust the naming accordingly.
- Appendix K: We add a new case study analyzing the distributions of the base LM, KNN-LM, and the MLP memory.
- Appendix L: We provide additional architectural details of the MLP to make the design clearer.
- Appendix M: We include an ablation study examining how different MLP memory sizes affect performance.

We hope these revisions improve the clarity and completeness of our submission.

---

### Author Response · Authors · 2025-11-30
**Summary of Rebuttal (Part 1/2)**

Dear Area Chair,

Thank you for taking over the assessment of our submission under the special circumstance. We truly appreciate your additional effort in evaluating the paper under this procedure. To facilitate your quick understanding of the paper, we provide a concise summary below.

> ## Summary of Strengths Highlighted by Reviewers

We thank all the reviewers **[R1 (xCCs), R2 (hdzS), R3 (qhzi), R4 (t9hD)]** for consistently acknowledging several strengths of our work across the initial reviews.

- They found the core idea of using an MLP-based parametric memory to imitate retriever‘s distribution both novel, intuitive, and clearly presented **[R1, R2, R3, R4]**.

- Reviewers appreciated that our approach substantially reduces storage requirements and removes retrieval overhead, leading to more efficient inference **[R1, R2, R3, R4]**.

- They further noted our strong empirical gains over base models and competitive baselines on factual QA, hallucination reduction, and general NLP benchmarks **[R2, R3, R4]**.

- Finally, reviewers highlighted the breadth of our evaluation and the informative ablations that clarify key design choices such as interpolation, attachment depth, and training objectives **[R1, R2, R3, R4]**.

> ## Key Concerns Raised and Our Responses

Below we summarize the main issues raised by each reviewer and briefly indicate how we addressed them. Detailed responses can be found in the rebuttal section.

> ### 1. New Main Experiments
- **[R1-W2] kNN-LM QA Performance (baseline)**

We added full QA comparisons showing that kNN-LM brings only modest improvements over the base LM, while MLP Memory delivers substantial gains across all QA benchmarks.

- **[R1-W3] Perplexity Evaluation**

We now report perplexity for the base LM, kNN-LM, and MLP Memory, and MLP Memory matches or surpasses kNN-LM, especially on Mistral.

- **[R3-Q3] Generalization to Specialized Domains**

Additional experiments on legal datasets (CaseHOLD, SCOTUS) show consistent improvements over both base LM and kNN-LM, indicating good transfer to specialized domains.

- **[R4-W2] Query Decomposition Performance (baseline)**

We add Least-to-Most prompting as a baseline and show that MLP Memory is better on most QA datasets.

- **[R4-W3]End-to-end RAG Comparison (baseline)**

We compare against vanilla RAG and REPLUG under a zero-shot setup, finding that MLP Memory outperforms REPLUG on most datasets while RAG/REPLUG retain advantages on a few tasks.

---

> ### 2. New Ablation and Case Study Analyses
- **[R1-W4, R3-W3] Distribution Analysis**

We hypothesize that kNN distributions excel at long-tail facts while LM distributions preserve coherence, and case studies show that MLP Memory inherits both behaviors (kNN-like on rare entities, LM-like on function words).

- **[R2-W4&Q4] Sensitivity of interpolation weight λ**

We provide a sensitivity study (Appendix D) showing that performance is stable for λ at roughly 0.3–0.6.

- **[R2-Q5, R4-Q2] Scaling of MLP Memory**

Ablations on small/medium/large MLPs (Appendix M) show that even the smallest configuration brings large gains, with diminishing returns from further scaling, and an 8-layer (~1B) configuration offering the best trade-off.

- **[R3-W2&Q2] Knowledge Updating**

Sequential training on disjoint Wikipedia shards improves perplexity and QA performance without catastrophic forgetting, showing that MLP Memory can be incrementally updated through further training.

- **[R4-Q3] Decoder-like Memory**

We compare to a decoder-style memory of the same size and show MLP Memory performs better under the same objective, suggesting that an MLP architecture is better suited for directly mapping a single hidden state to a vocabulary distribution without extra token-mixing attention.

---

> ### Author Response · Authors · 2025-11-30
> **Summary of Rebuttal (Part 2/2)**
>
> > ### 3. Methodology and Design Clarifications
>
> - **[R1-W1] Limited Contribution**
>
> We clarify that MLP Memory is a shift from non-parametric retrieval to a parametric memory trained via retrieval imitation and next-token prediction, enabling end-to-end optimization and yielding much larger gains and efficiency than kNN-LM.
>
> - **[R2-W1&Q3, R3-W1&Q1] Quantification of training cost**
>
> On an 8×A800×8-node setup, the total MLP Memory training cost (4h LM inference, 8h kNN search, 12h MLP training) equals merely 2 epochs of CPT for the 7B model, which we believe is reasonable given the substantial performance gains achieved.
>
> - **[R2-W2&Q2] Base model frozen or jointly trained**
>
> We keep the base LM frozen during MLP Memory pretraining and view joint training as future work.
>
> - **[R2-W3] Plug-and-play capabilities**
>
> We pretrain one MLP Memory per base LM on a general corpus (e.g., full Wikipedia) and reuse it across all tasks relying on that corpus, similar in spirit to RAG but with a separate memory for each model.
>
> - **[R2-W5] Optimal attachment depth**
>
> We show that attaching MLP Memory at 70–80% of depth is consistently near-optimal across architectures, and performance degrades only mildly beyond that (Section 5.4, paragraph 3).
>
> - **[R2-Q1] Generalization ability**
>
> We demonstrate that a single MLP Memory trained on 5B-token Wikipedia generalizes across all Wikipedia-based tasks, and discuss scaling to larger pretraining corpora as future work.
>
> - **[R2-Q6] Related literature**
>
> We compare MLP Memory with Product Key Memory methods, Entities as Experts, and UltraMemory, highlighting that our dense memory differs in memory usage, training stability, and fully self-supervised pre-training without annotations, and we present it as a new dense memory paradigm considered alongside existing sparse architectures.
>
> - **[R3-W4] End-to-end RAG Systems**
>
> We will cite RetroLLM in the related work and clarify that, while both pursue unified retrieval–generation via different approaches, MLP Memory uses a simpler fully parametric training pipeline and single-pass inference with better efficiency, whereas RetroLLM focuses on transparent evidence generation.
>
> - **[R4-Q1] Dataset construction**
>
> We clarify that Figure 4 is illustrative and that, in practice, we construct embedding–distribution pairs for every token in the pretraining corpus, making the pipeline fully self-supervised.
>
> ---
>
> > ### 4. Addressing Presentation Issues and Missing Details
> - **[R4-W1] Presentation Issues**
>
> We clarify these points in the revised PDF, specifically in Section 4 (Implementation Details) and Appendix L, where we describe the 8-layer (~1B-parameter) MLP Memory configuration, selected based on our scaling experiments ([R2-Q5, R4-Q2]) as providing the optimal balance between performance and efficiency, and justify the choice of the Wikipedia-2021 dump by following the Atlas setup.
>
> ---
>
> We appreciate your time and effort in evaluating our submission, and we are happy to provide further clarification if anything is unclear.

---

### Meta-Review · Area_Chair_aHvg · 2025-12-30

**Summary:**

This submission proposes MLP Memory, a lightweight parametric memory module that is pretrained to imitate a kNN-LM retriever distribution over a large corpus, then interpolated with a frozen base LM’s next-token probabilities at inference. The goal is to bridge the latency and integration drawbacks of RAG-style retrieval with the forgetting risks of parametric fine-tuning. Across multiple QA, hallucination, and general NLP benchmarks, the paper reports consistent gains over the base LM and strong efficiency improvements versus retrieval-heavy baselines, supported by ablations on attachment depth, interpolation weight, objectives, and memory scale.

The rebuttal substantially improves the paper by adding missing comparisons and clarifications, including kNN-LM QA and perplexity results, clearer training details and compute reporting, sensitivity and scaling ablations, broader baselines such as Least-to-Most prompting and zero-shot RAG/REPLUG, and additional experiments on legal datasets that suggest some transfer beyond Wikipedia-style corpora. However, key concerns remain: the practical scope and “plug-and-play” claim are still not fully convincing, as the approach appears to require a separate memory per base model and potentially re-pretraining for new corpora, limiting universality relative to standard retrieval systems; retrieval baselines, while improved, could be stronger and more standardized, and some tasks still favor RAG-style methods; and the paper lacks a crisp articulation of what is fundamentally new beyond distilling retrieval into parameters, with limited mechanistic or theoretical insight into when the method should succeed or fail. Overall, the work shows promise but retains notable gaps, leading to a borderline accept decision.

**Reviewer Concerns:**

The rebuttal adds several missing comparisons and clarifications. It now includes kNN-LM QA results and perplexity comparisons, helping evaluate whether the MLP approximates retrieval behavior. It also clarifies the training pipeline and provides a GPU-hour breakdown, specifies that the base model is frozen during memory pretraining, adds λ sensitivity and memory scaling ablations, and expands baseline coverage with Least-to-Most prompting and a zero-shot RAG/REPLUG comparison. Additional experiments on legal datasets provide some evidence of transfer beyond Wikipedia-style corpora.

Despite these additions, some concerns remain.

The method’s practical scope and “plug-and-play” claim are still not convincingly supported. While the authors clarify that a memory trained on a corpus can generalize across tasks drawing from that corpus, the approach still appears to require one memory per base model (and potentially re-pretraining for substantially different corpora), which limits universality compared to retrieval systems that can swap indices without retraining. Additionally, the added RAG/REPLUG comparisons are useful, but there is still room for stronger and more standardized end-to-end retrieval baselines and/or clearer discussion of what is feasible under comparable compute. In particular, some tasks still show RAG-style approaches retaining advantages, and the framing could be more explicit about when MLP Memory is expected to win.

A minor writing issue is that the paper still lacks a crisper articulation of what is fundamentally new beyond “distilling retrieval into parameters.” The additional case studies help, but they do not yet yield a compelling mechanistic or theoretical account of when and why this should work reliably, or what its failure modes are.

**Reviewer Scores:**

Reviewer xCCs (initial rating 4, confidence 2): Maybe improve to 6. The major missing item (kNN-LM QA baseline) and perplexity comparisons were supplied, and the distribution analysis rationale was strengthened with case studies.

Reviewer hdzS (initial rating 6, confidence 4): Stay at 6. The rebuttal directly addresses training cost, frozen-backbone detail, λ sensitivity, scaling, and adds substantial related-work discussion. Remaining concerns are more about generality and positioning than correctness.

Reviewer qhzi (initial rating 4, confidence 4): Maybe improve to 6. The authors addressed compute cost, demonstrated incremental updates, added specialized-domain experiments, and agreed to incorporate RetroLLM in related work; these collectively reduce the main objections.

Reviewer t9hD (initial rating 6, confidence 4): Likely no change. Presentation details (MLP layers, corpus choice, dataset construction clarification) and new baselines (Least-to-Most; RAG/REPLUG) resolve most clarity and comparison concerns, though the reviewer’s skepticism about absolute task performance and whether compute is best spent elsewhere may remain.

---

### Decision · Program_Chairs · 2026-01-26

Accept (Poster)